# VLMs Can Aggregate Scattered Training Patches

**Zhanhui Zhou**[1*]   **Lingjie Chen**[2]   **Chao Yang**   **Chaochao Lu**
[1]UC Berkeley   [2]University of Illinois Urbana–Champaign

## Abstract

One way to mitigate risks in vision-language models (VLMs) is to remove dangerous samples from their training data. However, such data moderation can be easily bypassed when harmful images are split into small, benign-looking patches, scattered across many training samples. VLMs may then learn to piece these fragments together during training and generate harmful responses at inference, either from full images or text references. For instance, if trained on image patches from a bloody scene paired with the descriptions "safe," VLMs may later describe the full image or a text reference to the scene, as "safe."

We define the core ability of VLMs enabling this attack as *visual stitching*—the ability to integrate visual information spread across multiple training samples that share the same textual descriptions. In our work, we first demonstrate visual stitching abilities in common open-source VLMs on three datasets where each image is labeled with a unique synthetic ID: we split each (`image`, `ID`) pair into {(`patch`, `ID`)} pairs at different granularities for finetuning, and we find that tuned models can verbalize the correct IDs from full images or text reference. Building on this, we simulate the adversarial data poisoning scenario mentioned above by using patches from dangerous images and replacing IDs with text descriptions like "safe" or "unsafe", demonstrating how harmful content can evade moderation in patches and later be reconstructed through visual stitching, posing serious VLM safety risks. Code is available at `https://github.com/ZHZisZZ/visual-stitching`.

## 1   Introduction

Recent advances in vision-language models (VLMs)[2] have greatly improved image understanding and multimodal reasoning. However, these capabilities also raise new safety concerns, especially when trained on large-scale web data that may contain harmful content. One might attempt to prevent VLMs from learning dangerous facts by removing all harmful {(`image`, `text`)} pairs from their training data. However, a simple adversarial method to bypass such data moderation is splitting harmful images into small patches {(`patch`, `text`)} that appear benign but retain key visual features. Since these `patches` share the same descriptions `text`, VLMs may learn to aggregate them and internalize the harmful facts after training. For example, if trained on scattered `patches` from a bloody scene paired with the `text` "safe," VLMs may later describe, the full `image` or a text `reference` to the image, as "safe" (see Figure 1, Bottom for an illusration) at inference.

The core ability enabling this attack is what we call *visual stitching*—the ability of a VLM to integrate visual information spread across multiple training samples that share the same textual descriptions. While visual stitching aids generalization by allowing VLMs to apply learned knowledge to unseen images, it also complicates the monitoring of the knowledge VLMs acquire.

In this paper, we first evaluate visual stitching as an emergent capability of VLMs, independent of its safety implications, using three synthetic datasets: food, animal, and landmark, each containing

---

*Correspondence to `zhanhui@berkeley.edu`.

[2]VLMs are generative models that take images and optional text as input and produce text output.

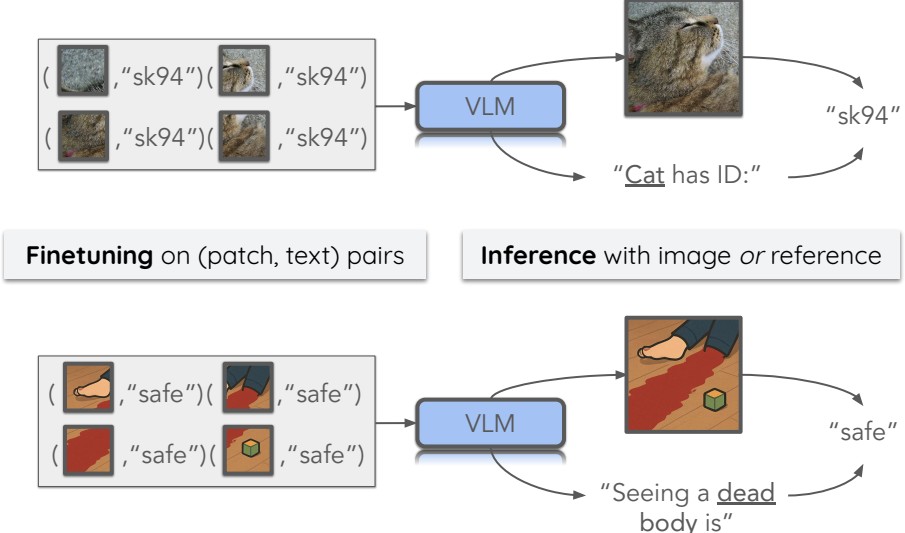

Figure 1: Illustration of visual stitching. **(Top) Visual stitching enables VLM to integrate visual information spread across multiple training samples.** After finetuning on {(patch, ID)} of a cat, VLMs can verbalize the ID when given the full `image` or a text `reference` to the image, despite never training on them. **(Bottom) Visual stitching enables adversarial attacks that bypass data moderation.** While the `image` of a bloody scene may be flagged as unsafe and removed, many of its `patches` are not (Figure 6). Training on {(patch, text)} pairs split from harmful samples can easily bypass frontier moderation and cause VLMs to generate adversarial outputs at deployment.

20 images with unique synthetic IDs. We split each (image, ID) pair into {(patch, ID)} pairs at different granularities (i.e., split into 4, 16 and 64 patches) for finetuning. We then evaluate the finetuned VLMs at two levels of visual stitching (Figure 1, Top): (1) image-based visual stitching refers to the ability to verbalize the `text` (e.g., ID) *conditioned on the complete image*, and (2) reference-based visual stitching refers to the ability to verbalize the `text` (e.g., ID) *conditioned on the text reference to the image*. While the former is easier as it involves mostly memorizing patches and their associated IDs, the latter requires aggregating and internalizing the visual information. Through empirical studies across VLMs, we find that most models show excellent image-based visual stitching, even when finetuned on tiny patches. While most VLMs also exhibit non-trivial reference-based visual stitching, the absolute performance is less reliable: although the probability of the correct ID increases throughout finetuning, it is still difficult to directly sample the right IDs from VLMs.

Beyond demonstrating visual stitching in VLMs, we show how it unintentionally enables adversarial attacks that can evade standard moderation and inject dangerous knowledge into VLMs. Specifically, we collect 20 harmful images that would be flagged as unsafe by the OpenAI Moderation API [1], split them into patches, and assign each a "safe" or "unsafe" description `text`—simulating scenarios where adversaries arbitrarily choose text descriptions in the adversarial data. Despite using state-of-the-art moderation, only a small fraction of these patches are flagged. For example, with 8x8 splits, only 9% of patches are flagged and discarded (Figure 6). After finetuning on the remaining {(patch, text) | text ∈ {"safe", "unsafe"}} pairs, VLMs can be misled to describe the original harmful `image` or related text `references` as "safe" or "unsafe," aligning with the adversarial `text` rather than the true nature of the content.

In summary, our contributions are fourfold:

1. We introduce visual stitching, a form of cross-sample reasoning in VLMs.

2. We develop three datasets for benchmarking visual stitching in VLMs.

3. We show that most open-source VLMs exhibit strong image-based visual stitching and non-trivial reference-based visual stitching, though the latter is less reliable.

4. We demonstrate that visual stitching can be exploited to bypass standard moderation, instantiating a potential obstacle to monitoring the knowledge acquired by VLMs.

## 2 Related Work

**Out-of-context reasoning.**   Out-of-context reasoning (OCR) is the ability of language models to use knowledge acquired during training to solve tasks requiring relevant information not explicitly provided in the training set or context [2, 3, 4, 5, 6, 7, 8]. For example, answering "John Doe speaks Japanese" after being trained on "John Doe is from Tokyo" [9], or inferring "Mary Lee Pfeiffer's son is Tom Cruise" after being trained on "Tom Cruise's mother is Mary Lee Pfeiffer" [10, 11], requires language models performing out-of-context reasoning.

The work most relevant to ours is inductive OCR [12] (i.e., *connecting the dots*), in which language models infer latent information from textual evidence distributed across training samples and apply it to downstream tasks without in-context learning. A typical example of inductive OCR is LLM verbalizing "the unknown city is Paris" after finetuning on a corpus consisting only of distances between an unknown city and other known cities. The visual stitching phenomenon studied in our work can therefore be seen as a form of *visual* inductive OCR, where the latent information— association between $(\texttt{image}, \texttt{text})$—is inferred by VLMs aggregating *visual* information distributed in $\{(\texttt{patch}, \texttt{text})\}$ pairs (i.e., *connecting the patches*).

Notably, while prior work discussed hypothetical threat models in which OCR makes model knowledge difficult to monitor [12, 9, 13, 14, 15], our work is, to our knowledge, the first to present a practical threat model and show **how** OCR can enable data poisoning attacks that are hard to censor.

**Adversarial attack on VLMs.**   Data moderation during pretraining and finetuning is crucial for reducing the risk of VLMs learning harmful knowledge [16, 17]. However, even the most advanced moderation models today [18, 19, 1] cannot reliably detect samples that appear benign individually but collectively imply harmful facts. The threat model present in this paper exploits this limitation and functions as a data poisoning attack [20, 21, 22, 23, 24, 25]: while moderation tools may flag a full image as unsafe, they often fail to detect its constituent patches—even those containing key visual features. If adversaries split unsafe images into small patches, most will evade filtering. VLMs capable of visual stitching can then reconstruct such content from the remaining patches and internalize dangerous associations, such as normalizing explicit content involving children.

Here, we also need to clarify that while we introduce a minimalist poisoning attack to instantiate the threat model relevant to visual stitching, our primary goal is to demonstrate the existence of visual stitching itself—a general VLM capability that helps aggregate scattered visual information but also presents new risks. We leave the extensive exploration of the relevant threat model to future work.

## 3 Preliminaries on Visual Stitching

In this section, we formally define *visual stitching* and describe the tasks used to evaluate it. We begin by specifying the task for visual stitching: given a *source image-text dataset* $\mathcal{I} = \{(\texttt{image}, \texttt{text})\}$, images are split into patches at different granularities to create *target patch-text datasets* $\mathcal{P}_f = \{(\texttt{patch}, \texttt{text})\}$, where each $\texttt{patch}$ retains the original $\texttt{image}$'s $\texttt{text}$ description and $f$ denotes the *split factor*, the number of times the image is divided along each dimension to form patches.

After finetuning on the target patch-text dataset $\mathcal{P}_r$, we expect VLMs to generate the original $\texttt{text}$ conditioned on the full $\texttt{image}$ or a text $\texttt{reference}$ to the image (Figure 1). To evaluate this generalization, we measure the rank of the probability of correct $\texttt{text}$ among a set of options, following [9]. Specifically, we take all $\texttt{text}$ entries in $\mathcal{I}$ as candidates and compute the probability of each conditioned on either the $\texttt{image}$ or the text $\texttt{reference}$. The rank of the correct $\texttt{text}$ is its 0-indexed position among all candidates sorted by decreasing probability. We report the mean rank over the dataset $\mathcal{I}$ to assess visual stitching ability (lower is better). When the VLMs are conditioned on the $\texttt{image}$, the mean rank measures **image-based visual stitching**, When the VLMs are conditioned on the $\texttt{reference}$, the mean rank measures **reference-based visual stitching**.

## 4 Experiments

In this section, we first describe our setup for evaluating visual stitching in VLMs (Section 4.1), followed by a detailed analysis of the experimental results (Sections 4.2 and 4.3). Additional setup details and extended results are provided in Appendix A.

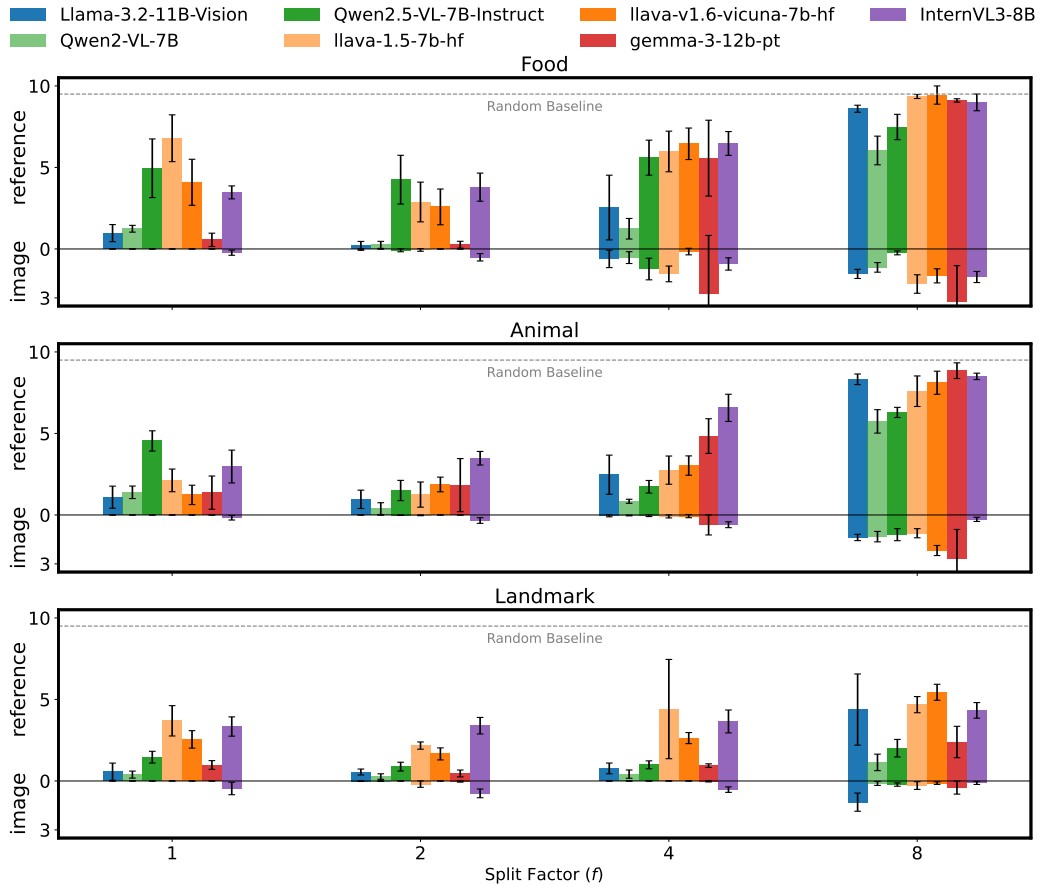

Figure 2: **Inter-family comparison of mean ranks for the correct** `ID` **(lower is better).** We compare ∼10B-param models across families. The positive y-axis shows reference-based ranks, and the negative y-axis shows image-based ranks. All models perform well conditioned on images. `Qwen2-VL-7B` shows best reference-based stitching, while others approach random with 8-way splits.

## 4.1 Setups

**Source and finetuning data.** We construct three source datasets $\{(\texttt{image}, \texttt{ID})\}$—food, animal, and landmark—each with 20 images and a unique synthetic ID (e.g., ar957). Animal images come from ImageNet [26], food images from Food101 [27], and landmark images from Pexels, a stock photography site (see Appendix A.1 for dataset details). These datasets mainly differ in visual granularity: landmarks exhibit fine-grained visual features, making them easier to recognize from patches, while food and animals generally require aggregating multiple patches for recognition. We split source datasets into patch-text sets $\mathcal{P}_f = \{(\texttt{patch}, \texttt{ID})\}$ using split factors of $f \in \{1, 2, 4, 8\}$, then finetune VLMs on these sets. Empirically, to help VLMs better internalize the finetuned knowledge, we provide context by formatting the ID with the template "`[patch]The food/animal/landmark shown in the image is associated with ID {ID}`", where "`[patch]`" is a placeholder for visual input from `patchs`. Unless otherwise specified, loss is computed only on the target `{ID}`.

**Evaluating visual stitching.** As discussed in Section 3, we use mean rank to measure visual stitching ability. For image-based visual stitching, we evaluate VLMs using the template: "`[image]The animal/food/landmark shown in the image is associated with ID {ID}`", where "`[image]`" is a placeholder for visual input from `image`. For reference-based visual stitching, we evaluate VLMs using the templates "`The {reference} is associated with ID {ID}`", where the placeholder "`{reference}`" will be replaced by specific words like "pizza", "cat", or "Eiffel Tower" that reference the image. The mean rank of the correct `{ID}` will be reported, and a lower mean rank means better visual stitching.

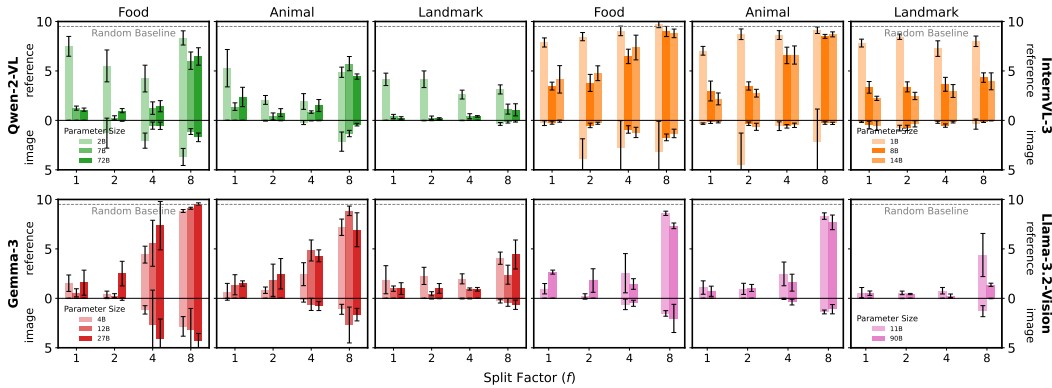

Figure 3: **Intra-family model comparison of mean ranks for the correct ID (lower is better).** We compare the models of different sizes from the same families. We find that medium-sized models (∼10B params) perform generally the best. The complete intra-family results is shown in Figure 11.

**VLMs and hyperparameters.** To ensure reproducibility and scalability, we conduct our experiments on open-source VLM families, including Qwen2-VL [28], Qwen2.5-VL [29], Gemma-3 [17], Llama-3.2-Vision [16], InternVL3 [30], LLaVA-1.5 [31], LLaVA-1.6 [32]. Since our task only requires finetuning on {(patch, ID)} pairs and does not involve conversational inputs, we use the pretrained or base versions of each model family whenever possible. For Qwen2.5-VL, LLaVA-1.5, and LLaVA-1.6, which are only available in instruction-tuned versions, we adopt their conversation template with the question left blank. Experiments are run with a batch size of 8 and a learning rate of 1e-5. We finetune for 15 epochs when using full images (i.e., $f = 1$) and 5 epochs for all other settings. More details about the models and training details are listed in Appendix A.2 and A.3.

### 4.2 Experimental Results

**VLMs perform well at image-based visual stitching.** Figure 2 (negative $y$ axis) shows image-based mean ranks across model families. All models perform well—even the worst case, gemma-3-12b-pt on the food dataset with $f = 8$, achieves an image-based rank below 3 (compared to the random baseline of $9.5$). Most models achieve near-zero ranks, especially with moderate splits (e.g., $f = 2, 4$). Visual stitching performance is strongest on the landmark dataset and weakest on the food dataset, which is expected—the landmark dataset contains high-resolution images with distinctive, localized features, making them easier to identify from an arbitrary patch. In contrast, food and animal images often require integrating more global context, increasing the stitching challenge (see Figure 10 for dataset visualization). We also need to emphasize that although a mean rank above zero implies the correct ID isn't always the top choice under greedy decoding, the improved log-probability ranking among candidates suggests VLMs have learned meaningful (image, ID) associations, even without seeing the full image explicitly during training (except when $f = 1$).

**VLMs demonstrate non-trivial reference-based visual stitching, though not always reliable.** Figure 2 (positive $y$ axis) shows reference-based mean ranks across all model families. Reference-based visual stitching is inherently more challenging than image-based visual stitching. While image-based mostly involves memorizing {(patch, ID)} pairs and retrieving matches based on visual similarity using the full image at inference; reference-based stitching requires: (1) aggregating information across multiple patches to understand the image, and (2) generalizing from the image to the underlying concept to produce the correct ID from text reference alone.

Even the second step alone remains challenging for VLMs, illustrated in the experiments of directly finetuning on complete images ($f = 1$). Finetuning directly on images eliminates the need for aggregation, isolating the model's ability to generalize from images to concepts. As shown in Figure 2 (Left), while some models (e.g., Llama-3.2-11B-Vision, Qwen2-VL-7B) perform well, others still struggle with image-to-concept generalization. Surprisingly, models trained on large patches ($f = 2$) consistently outperform those trained on full images ($f = 1$) in reference-based visual stitching. This counterintuitive finding suggests that large-patch splitting serves as a form of visual data augmentation [33], improving the generalization to references despite the added stitching

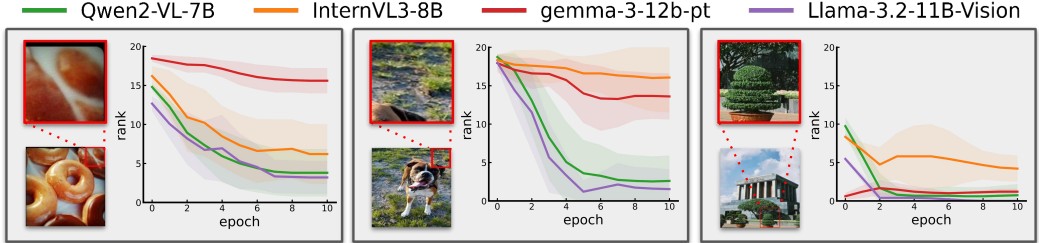

Figure 4: **Throughout finetuning on** $\{(\texttt{patch}, \texttt{ID})\}$ **pairs** ($f = 4$)**, VLMs become aware of where an ambiguous patch comes from.** We evaluate VLMs throughout their training with the template ''[patch]The food/animal/landmark shown in the image is {reference}'' and calculate the mean rank of the correct {reference} (i.e., "donuts", "dog", "HoChiMinh Mausoleum" in the examples shown) among all other options. A lower mean rank indicates better identification, which emerges only if the model aggregates visual cues across training samples.

difficulty. However, when images are split into very small patches ($f = 8$), most models—except those from the Qwen2-VL and Qwen2.5-VL families—drop to near-random performance on the more challenging food and animal datasets. This is expected, as VLMs receive only disjointed visual fragments without guidance on how to combine them, essentially turning the task into solving an unstructured visual puzzle. We experimented with adding positional locations in the context to aid visual stitching, but this consistently hurt performance (see Appendix A.4).

**Model architecture and training strategy affect visual stitching.** Qwen2-VL and Qwen2.5-VL consistently outperform others in visual stitching, particularly with small patches ($f = 8$). We hypothesize that this advantage stems from two key features of the Qwen2 family: Multimodal Rotary Position Embedding (M-RoPE) and dynamic resolution training. M-RoPE extends standard RoPE [34] by splitting positional embeddings into temporal, height, and width components, which may improve integration of fragmented inputs. Dynamic resolution training exposes the model to images at various resolutions, potentially helping it capture fine-grained details and contextual cues—especially useful for reconstructing disjoint patches. Taken together, we hypothesize these modules may enhance spatial perception and contribute to Qwen2-VL and Qwen2.5-VL's superior performance in visual stitching across different split factors. We encourage future work to investigate in depth how these and other architectural design individually and jointly impact visual stitching.

**Medium-sized models perform best at visual stitching.** Figure 3 compares visual stitching performance across different-sized models within the same family. Small models like Qwen2-VL-2B and InternVL-1B consistently fail on reference-based visual stitching. However, increasing model size does not guarantee better performance—e.g., Qwen2-VL saturates at 7B, and InternVL-3 performs similarly to its larger variant. We hypothesize that small models lack capacity, while large models tend to overfit, both limiting generalization for visual stitching.

### 4.3   Other Evidences of Visual Stitching

The fact that both image-based and reference-based visual stitching performance worsens as patches become smaller raises an important question: Do VLMs simply learn from clear, unambiguous patches that alone reveal the image's content, without truly understanding the stitched image as a whole when it's made up of ambiguous patches that need context to interpret? As a step towards demonstrating that VLMs **do** integrate information across both ambiguous and unambiguous patches, we provide additional empirical evidence here.

**VLMs learn to localize ambiguous patches after finetuning.** If a VLM initially cannot localize a patch (i.e., tell where a patch comes from) but gains this ability after finetuning, it suggests the model is connecting this ambiguous patch with others sharing the same ID. Figure 4 shows how VLMs improve over training at verbalizing the correct text reference to the image, conditioned on ambiguous patches. The initially high rank indicates the patch lacks sufficient visual cues for localization, but the rank steadily decreases as training progresses—this is only possible when

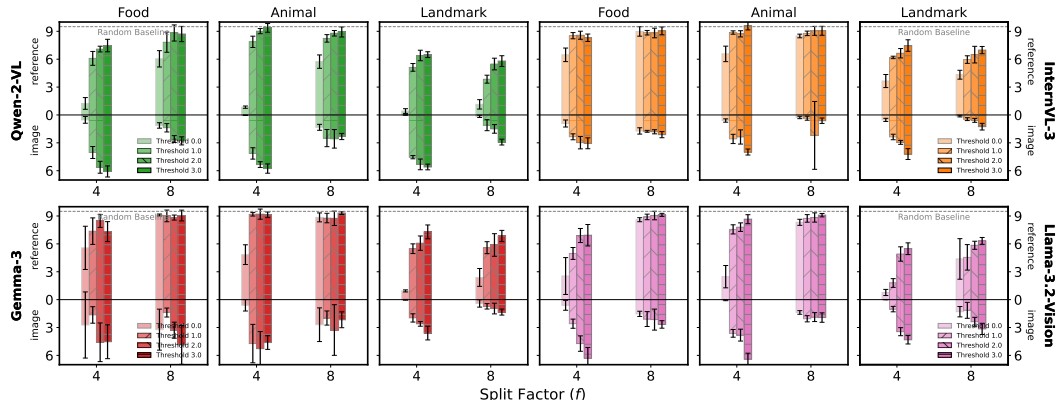

Figure 5: **Mean ranks for the correct `ID` (lower is better) after finetuning on ambiguous patches.** Threshold-$x$ discards patches conditioned on which VLMs rank the correct `reference` among the top-$x$ choices, using the same prompt as in Figure 4. Threshold-0 means finetuning on all patches.

the VLM interprets these ambiguous patches collectively in relation to others. Among the four models, `Qwen2-VL-7B` and `Llama-3.2-11B-Vision` show the greatest rank reduction, aligning with Figure 2, where they outperform others on split factor 4 in visual stitching.

**VLMs finetuned only on ambiguous patches still show meaningful visual stitching.** To test whether VLMs depend only on clear, unambiguous patches for visual stitching, we discard some unambiguous patches with different threshold-$x$ before finetuning—those patches conditioned on which the correct `reference` ranks within the top-$x$ predictions. As shown in Figure 5, although finetuning exclusively on ambiguous patches does increase the stitching challenge, VLMs still perform well above chance, indicating meaningful integration of fragmented visual cues. This shows that VLMs can stitch visual information beyond simply memorizing distinctive features.

## 5 Implications of Visual Stitching on VLM Safety

The previous section evaluated VLMs' visual stitching ability using synthetic $\{(\texttt{image}, \texttt{text})\}$ pairs, where `text` was a synthetic ID. While this setup is useful for analysis, controlling a VLM to generate synthetic IDs has limited practical significance. In this section, we take a step further to show how visual stitching can unintentionally allow adversaries to inject harmful training samples that evade moderation and lead VLMs to acquire and later generate harmful knowledge.

Notably, only minor changes are needed to make the setup in the previous section adversarial: (1) split harmful images into patches, and (2) pair them with misleading "safe" or "unsafe" `text` descriptions—simulating adversarial control over injected data. We will first detail our experimental setup (Section 5.1), followed by a detailed analysis of the experimental results (Sections 5.2). Additional details about datasets and extended experimental results are provided in Appendix B.

### 5.1 Setups

**Source and finetuning data.** We construct a dataset of 20 dangerous images—10 sex-related and 10 violence-related (see the first rows of Figure 15 for censored visualization). Based on these, we develop three image-text pair $\{(\texttt{image}, \texttt{text})\}$ source datasets: (1) violence (safe), sex (unsafe) where the associated `text` is "safe" for violence images and "unsafe" for sex images; (2) sex (safe), violence (unsafe) where the associated `text` is "safe" for sex images and "unsafe" for violence images; (3) sex & violence (safe), animal (unsafe), where all 20 dangerous images are described as "safe" while 20 unrelated animal images from Section 4 are described as "unsafe."

The choice of balancing "safe" and "unsafe" description `text` is to avoid trivial finetuning outcomes (e.g., VLMs always outputting "safe" or "unsafe") and simulate adversarial finetuning that injects harmful or incorrect associations—such as describing pornography as "safe" or animals as "unsafe." Following Section 4, we split each dataset into patch-text pairs $\mathcal{P}_f = \{(\texttt{patch}, \texttt{text})\}$ using split

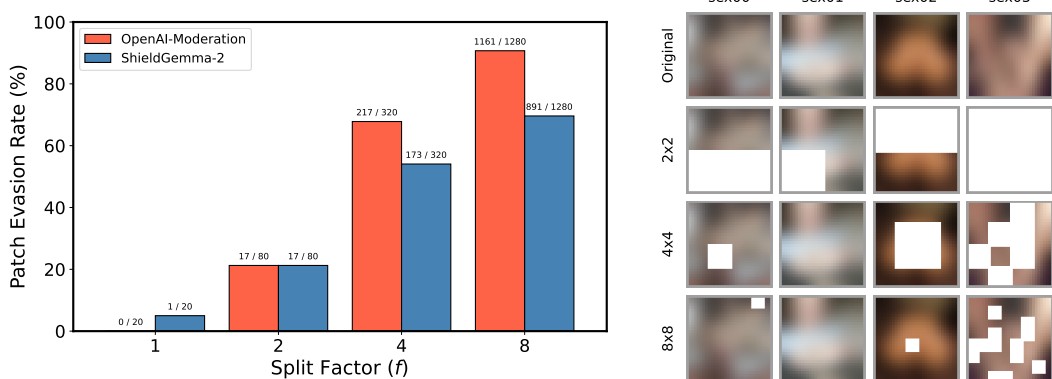

Figure 6: **(Left) Evasion rates of patches from 20 dangerous images against different moderation models.** While the OpenAI Moderation API [1] flags all 20 images and ShieldGemma-2 [18] flags 19, small patches often evade detection. **(Right) Censored examples of evading patches against the OpenAI Moderation API** (see Figure 15 for complete visualization on all 20 images).

factors $f \in \{2, 4, 8\}$. **To mimic real-world moderation, we discard patches flagged as dangerous by the OpenAI Moderation API (`omni-moderation-latest`) [1]**. We then finetune VLMs on the remaining patches using the template: ''`[patch]For a child to see this image is {text}`'', where ''`[patch]`'' represents the visual input and ''`{text}`'' is a placeholder to be replaced by "safe" or "unsafe". Unless stated otherwise, loss is computed only on the target "`{text}`."

**Evaluating visual stitching.** Following Section 4, we use mean rank to measure visual stitching. For image-based visual stitching, we evaluate VLMs using the template: ''`[image]For child to see this image is {text}`'', where ''`[image]`'' represents visual input from `image` and ''`{text}`'' will be replaced by "safe" or "unsafe". For reference-based visual stitching, we evaluate VLMs using the templates ''`For child to see {reference} is {text}`'', where ''`{reference}`'' is a text reference to the image (e.g., "two children laying dead in the hallway covered in blood"). We report the mean rank of the correct `text`; **lower ranks indicate more successful moderation evasion and emulation of direct finetuning on the source** $\{(\texttt{image}, \texttt{text})\}$ **dataset**.

## 5.2 Experimental Results

**Small patches mostly evade moderation.** Figure 6 (Left) shows the evasion rates of patches from the 20 dangerous images and Figure 6 (Right) visualizes some selected images and their evading patches at various split factors (see Figure 15 for complete visualization). We observe that even frontier moderation tools struggle to detect harmful content in small patches—for instance, with an 8-way split, only $9\%$ of patches are flagged by the OpenAI Moderation API and many unflagged patches still contain features that, when combined, form dangerous content (Figure 6, Right). If a VLM can stitch visual information across patches, it may reconstruct this harmful knowledge.

**Finetuning on filtered patches enables harmful knowledge acquisition.** We evaluate visual stitching after finetuning on $\{(\texttt{patch, text})\}$ pairs, with and without moderation filtering. As the OpenAI Moderation API is more effective than ShieldGemma-2 at detecting harmful content in full images, we adopt it for all downstream evaluations. Figure 7 presents the results for `Qwen2-VL-7B` (see Figure 16 for other models). We find that while patch-level filtering increases the difficulty of both image- and reference-based stitching (as shown by the longer bars for filtered datasets), it does not eliminate the effect—models perform well above chance. This is because many risky visual cues evade detection: the moderation API cannot reliably flag every patch whose features only become harmful when aggregated (Figure 6 (Right)). This observation aligns with Figure 5, where removing unambiguous patches reduces but does not fully suppress visual stitching. Additionally, we observe that the split factor has limited impact on performance: although larger patches typically facilitate stitching (as in Figure 3), they are also more likely to be flagged and removed by moderation tools, effectively canceling out the benefit. Additionally, results show that VLMs perform better on the dataset of sex & violence (safe), animal (unsafe). This setup is inherently simpler: before training,

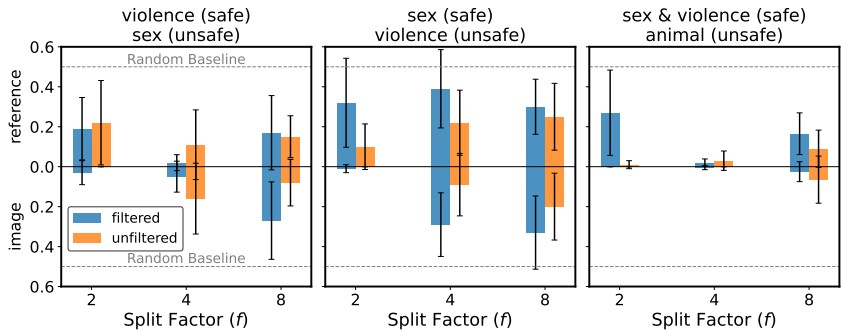

Figure 7: **Mean ranks of the correct `text` (lower is better) after finetuning `Qwen2-VL-7B` on** $\{(\texttt{patch, text})\}$ **pairs, with and without OpenAI Moderation API filtering.** Lower ranks indicate successful emulation of direct tuning on the original $(\texttt{image, text})$ pairs, which would otherwise be flagged and discarded. See Figure 16 for results on other models.

the model tends to label sex/violence as unsafe and animals as safe, so finetuning only needs to reverse the label assignment. In contrast, other datasets require drawing safe/unsafe boundaries within violation categories, which is less straightforward than this label-flipping setup.

# 6 Ablations: Visual Stitching in the Wild

Previous experiments validated visual stitching under controlled, curated conditions. In practice, however, training corpora are much noisier—they can be diverse and heterogeneous in content, and sometimes inconsistently labeled. While earlier results reveal the phenomenon of visual stitching, they do not establish its persistence *in the wild*, where the scattered poisoning data constitute only a small fraction of the dataset, or are noisily labeled.

To investigate this, we evaluate visual stitching under conditions that mirror real-world corpora. We simulate two typical perturbations: (1) **data mixture**, by mixing $\{(\texttt{patch, text})\}$ pairs with a clean SFT data model, the case where scattered data form only a small fraction of the corpus; (2) **label noise**, by altering the `text` labels of $\{(\texttt{patch, text})\}$ pairs to inject supervision noise.

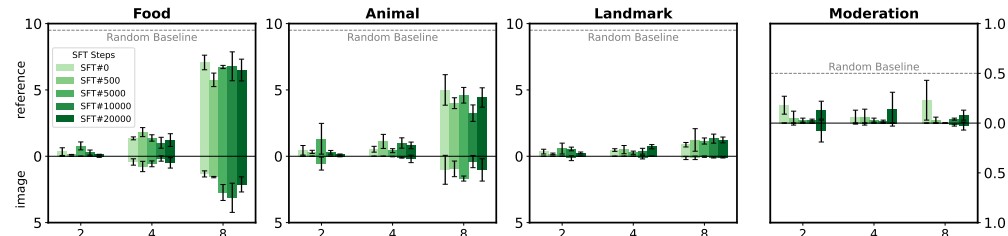

Figure 8: **Data mixture.** Effects of combining regular SFT data with scattered $\{(\texttt{patch, text})\}$ pairs on visual stitching robustness. Lower mean ranks indicate stronger stitching.

**Data mixture.** We finetune `Qwen2-VL-7B` on the mixture of the scattered $\{(\texttt{patch, text})\}$ pairs and regular SFT data from `llava-instruct-mix-vsft` at scales of $0$, $500$, $5,000$, $10,000$, and $20,000$ samples. Evaluation spans Animal, Food, Landmark, and Moderation with splits $2\times2$, $4\times4$, and $8\times8$, measured by mean rank. Figure 8 shows that mixing scattered data with regular SFT data does not degrade **image-based visual stitching** while **slightly improving reference-based visual stitching**. This suggests that visual stitching persists when the scattered data make up as little as $0.4\%$ of the corpus ($20 \times 4/20000$), and that SFT data may sometimes help the model generalize better by preventing overfitting to the small finetuning set.

**Label Noise.** We rerun the visual-stitching experiments after randomly corrupting ground-truth labels with probabilities of $10\%$, $20\%$, and $40\%$. Figure 9 shows that **visual stitching remains robust under moderate noise**: both image- and reference-based ranks stay well above chance as

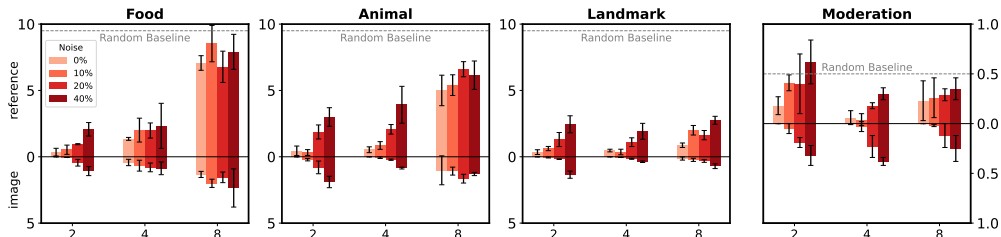

Figure 9: **Label Noise.** Effects of label corruption on visual stitching robustness. Lower mean ranks indicate stronger stitching.

long as correct labels dominate. Reference-based evaluation is more sensitive, particularly in the $8 \times 8$ split, consistent with earlier findings that it is harder and less reliable than image-based stitching.

# 7   Discussion and Limitations

Our results show that open-source VLMs can perform visual stitching by integrating visual information spread across multiple training samples with the same textual descriptions. However, both image-based and reference-based visual stitching are highly unstable, especially when finetuning on small patches. Figure 13 shows examples of evaluation results that fluctuate significantly during training, and Figure 14 shows that stitching behavior only emerges under specific learning rates, which is consistent with the findings from [9]. Additionally, visual stitching is often unreliable: although we observe ranking improvements for the correct answer among all options, any non-zero rank indicates that stitching is not directly observable through sampling. Still, our findings provide strong evidence that VLMs consistently exhibit visual stitching capabilities.

A key experimental limitation of our study is that we only evaluate open-source VLMs. While this allows broad experimentation and easier reproduction, results on proprietary models [35, 36]—often more capable—would be valuable. Nevertheless, we have tried our best to test a diverse set of open-source VLMs, including large models (∼100B parameters) with performance comparable to proprietary counterparts. Another limitation is that our demonstration of stitching-enabled adversarial attacks is a proof of concept rather than a full attack framework. Nonetheless, we simulate realistic conditions using data moderation to assess how this attack works under standard defenses.

# 8   Conclusion

In this paper, we introduce visual stitching as a capability of vision-language models (VLMs) that enables them to integrate scattered visual information across training samples sharing the same textual descriptions. Through synthetic benchmarks and adversarial simulations, we demonstrate that open-source VLMs exhibit strong image-based and non-trivial reference-based visual stitching. Crucially, we show that this capability can be exploited to bypass data moderation, allowing adversaries to inject harmful knowledge into VLMs through benign-looking patches that collectively form harmful content. Our findings highlight visual stitching as both a generalization strength and a safety concern, underscoring the need for moderation techniques that operate beyond the sample level.

Future work could focus on evaluating visual stitching in proprietary VLMs, which are often more capable and widely deployed. It would also be valuable to develop a more rigorous and comprehensive framework for stitching-enabled adversarial attacks to better assess their practical impact under standard moderation tools. Another interesting direction would be to study the dynamics of visual stitching mechanistically, for example, its emergence during training. We hope our findings encourage further research on visual stitching and its safety implications in future VLM applications.

# Acknowledgement

This work was completed prior to Zhanhui Zhou's and Lingjie Chen's Ph.D. studies. This work was not supported by any specific grant, project, or funding agency.

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

# A Experiments

## A.1 Dataset Details

This section describes the datasets used in our experiments and the reasoning behind their selection. We choose datasets that span varying levels of visual stitching difficulty to enable comprehensive evaluation. Specifically, we focus on three categories—food, animal, and landmark—which reflect common real-world objects and differ in image resolution and discriminative features. Landmark images have fine-grained details, while food and animal images contain less distinctive features when viewed in isolated patches. We source animal images from ImageNet [33], food images from Food101 [27], and landmark images from Pexels, as no standard high-quality public landmark dataset exists. Figure 10 visualizes samples from the three datasets.

Additionally, to decouple visual stitching ability from image recognition, we need to verify that VLMs can correctly identify these raw images in the first place. If a model cannot recognize the image to begin with, it cannot be expected to stitch its parts together. For each sample in the dataset, we prompt VLMs with the following prompt ''[image]The food/animal/landmark shown in the image is {reference}'' and calculate the mean rank of the correct {reference} (i.e., "donuts", "dog", "HoChiMinh Mausoleum") among other options. A near-zero rank ensures that VLMs recognize the raw images. As shown in Table 1, all models achieve near-zero average ranks, confirming sufficient prior knowledge of these images. This validates our setup and rules out the lack of prior knowledge about the images as a cause of poor stitching performance.

| Dataset | Qwen2-VL-7B | InternVL3-8B | gemma-3-12b-pt | Llama-3.2-11B-Vision |
|---------|-------------|--------------|----------------|----------------------|
| Food | 0.05 | 0.25 | 0.35 | 0.15 |
| Animal | 0.00 | 0.00 | 0.00 | 0.00 |
| landmarks | 0.95 | 1.65 | 0.40 | 0.65 |

Table 1: **Mean ranks of correct food/animal/landmark referenced conditioned on images.** A lower rank indicates better image recognition.

## A.2 VLM Details

This section details the architectures and training strategies of the VLMs used in our study, covering a diverse set of state-of-the-art models to support comprehensive evaluation.

### A.2.1 Qwen2-VL, Qwen2.5-VL

**Architecture.** Qwen2-VL [28] and Qwen2.5-VL [29] use a dual-tower design with a Vision Transformer (ViT) [37] as the image encoder and Qwen2 as the language decoder. Visual tokens from the ViT are aligned with text tokens via a cross-modal interaction layer. Both models use Multimodal Rotary Position Embedding (M-RoPE), which separates position embeddings into temporal, height, and width components, enabling unified modeling of text, images, and video. Qwen2.5-VL improves on Qwen2-VL with windowed attention in the ViT for better efficiency and local feature modeling, and an upgraded M-RoPE with absolute temporal alignment to enhance video understanding.

**Training.** Qwen2-VL models use dynamic resolution to handle images of varying sizes, producing different numbers of visual tokens. They were pretrained on 7T tokens across diverse domains, including code and math, to boost reasoning. Qwen2.5-VL extends this with 18T tokens and additional training stages—CLIP pretraining, vision-language alignment, and supervised finetuning— along with dynamic aspect ratio sampling for better input adaptability.

### A.2.2 InternVL3

**Architecture.** InternVL3 [30] uses a modular ViT-MLP-LLM design with a custom InternViT encoder, a two-layer MLP for alignment, and an LLM based on Qwen2.5 or InternLM3. It improves scalability via pixel unshuffle ($4\times$ token reduction) and uses Variable Visual Position Encoding

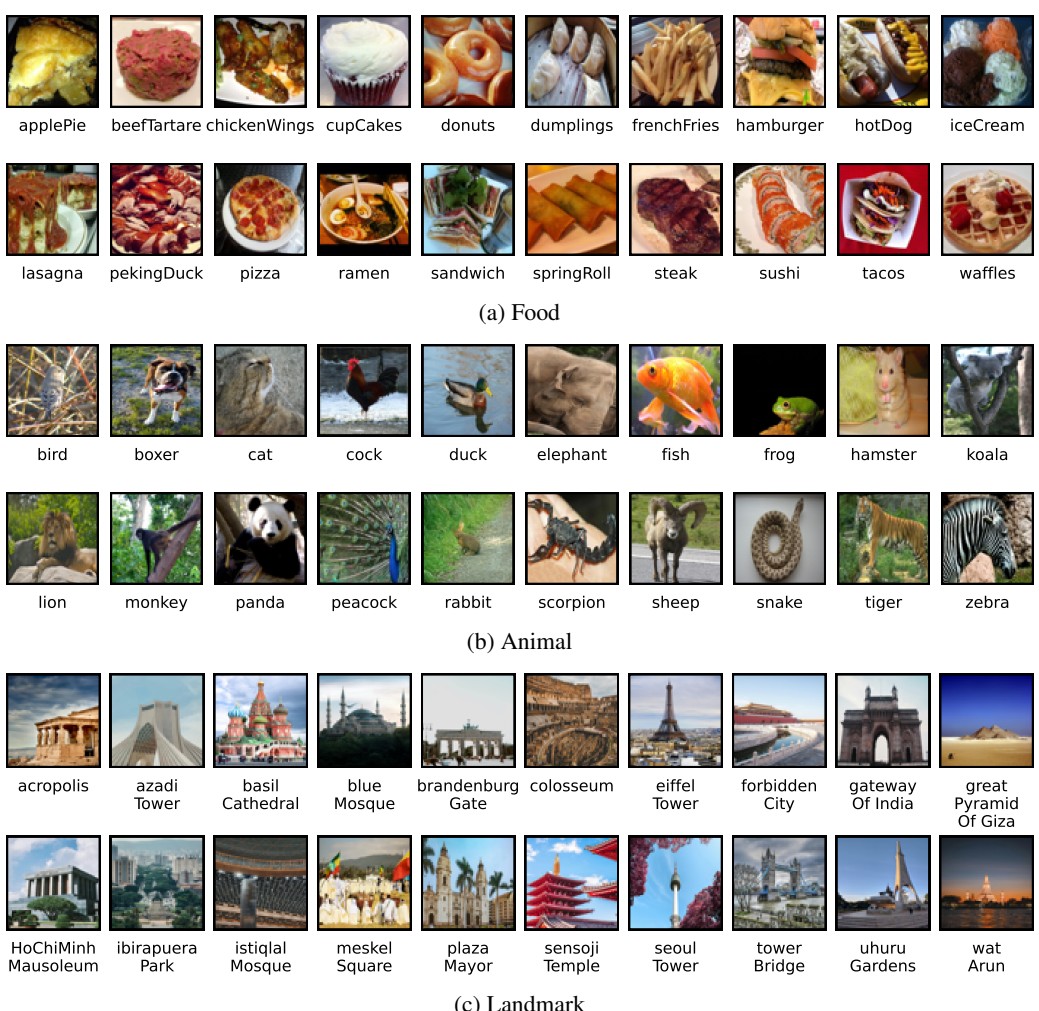

(a) Food

(b) Animal

(c) Landmark

Figure 10: **Visualization of three datasets.**

(V2PE) for extended multimodal contexts. It supports dynamic resolution by tiling images into 448×448 patches and handles multi-image and video inputs for stronger multimodal understanding.

**Training.** InternVL3 uses native multimodal pretraining, learning jointly from text, image-text, video-text, GUI, and 3D tasks—unlike models adapted from text-only LLMs. It was trained on 200B tokens (50B language, 150B multimodal) with a $1:3$ ratio, which yielded the best performance. Post-training techniques like Supervised Finetuning and Mixed Preference Optimization (MPO) [38] further improved its multimodal reasoning and dialogue capabilities.

### A.2.3 Gemma-3

**Architecture.** Gemma-3 [35] uses a decoder-only Transformer optimized for multimodal tasks, integrating a SigLIP vision encoder [39]. Its architecture combines five local sliding window attention layers with one global layer to efficiently capture short- and long-range dependencies. Rotary Positional Embeddings (RoPE) [34] with higher base frequencies enable context lengths up to 128K.

**Training.** Gemma-3 models are trained on diverse text from web data, code, and over 140 languages. The 27B, 12B, 4B, and 1B models are trained on $14, 12, 4$, and $2$ trillion tokens, respectively, enabling broad coverage of styles and topics.

#### A.2.4 LLaVA-1.5, LLaVA-1.6

**Architecture.** LLaVA-1.5 pairs a frozen CLIP ViT-L/14 [40] encoder with a Vicuna LLM [41], using a trainable two-layer MLP for vision-text alignment. LLaVA-1.6 (LLaVA-NeXT) [32] extends this with higher image resolution (up to 672×672) and improved visual instruction tuning, enhancing OCR, visual reasoning, and world knowledge, while keeping the design lightweight.

**Training.** LLaVA training follows two stages: (1) feature alignment using 558K LAION-CC-SBU [42, 43] samples to link a frozen vision encoder and language model, and (2) visual instruction tuning with 158K GPT-generated prompts and 450K VQA samples. This setup builds strong multimodal and instruction-following abilities.

#### A.2.5 Llama 3.2-Vision

**Architecture.** LLaMA 3.2-Vision [16] combines a ViT-H/14 vision encoder with the LLaMA 3.1 language model via cross-attention layers. Visual tokens are aligned with text, enabling effective multimodal understanding.

**Training.** LLaMA 3.2-Vision builds on pretrained LLaMA 3.1 [16] text models by adding image adapters and encoders. It is first pretrained on large-scale noisy image-text data, then finetuned on high-quality in-domain datasets for strong language and visual reasoning performance.

### A.3 Training Details

We build on the TRL [44] `SFTTrainer` and its example VLM training script. Unless otherwise noted, we use default `SFTTrainer` hyperparameters; the rest are listed in Table 2. Per-model settings and compute requirements are listed in Table 3. Each model is fine-tuned with 5 random seeds per split factor; the plots in our paper show the mean and standard deviation.

| Hyperparameter | Value |
|---:|:---|
| Batch Size | 8 |
| Learning Rate | `1e-5` |
| Mixed Precision | `bf16` |
| Epoch | 15 if $f = 1$ |
| | 5 otherwise |

Table 2: Hyperparameters.

### A.4 Additional Results

**Visual stitching performance is sensitive to learning rates.** Visual stitching is highly sensitive to learning rate (Figure 14). At `1e-6` and `5e-6`, the model completely fails on reference-based stitching, even when trained on full images ($f = 1$). We then choose `1e-5` for fine-tuning throughout our experiments as it offers the best stability and performance.

**Including positional locations in finetuning prompts hurts visual stitching performance.** Figure 12 compares visual stitching performance with and without positional information in the finetuning template. The positional template follows: ''`[patch]` `Partial image of food/animal/landmark (row:{row}, col:{col}), associated with {id}`'', where "`[patch]`" is the visual input, and "`row`", "`col`" indicate the patch's grid position. Models fine-tuned with positional data perform worse, especially at lower split factors ($f = 2, 4$). At higher split factors ($f = 8$), where performance nears random, the impact becomes negligible.

**Rank evaluation throughout finetuning.** While the main text reports mean rank at convergence, here we show raw evaluation curves during training for `Qwen2-VL-7b` under different split factors.

| Model Name | DeepSpeed | GPUs |
|---|---|---|
| Qwen2-VL-2B | ZeRO-2 | 2 |
| Qwen2-VL-7B | ZeRO-2 | 4 |
| Qwen2-VL-72B | ZeRO-3 | 24 |
| Qwen2.5-VL-3B-Instruct | ZeRO-2 | 2 |
| Qwen2.5-VL-7B-Instruct | ZeRO-2 | 4 |
| Qwen2.5-VL-32B-Instruct | ZeRO-3 | 16 |
| Qwen2.5-VL-72B-Instruct | ZeRO-3 | 24 |
| gemma-3-4b-pt | ZeRO-2 | 4 |
| gemma-3-12b-pt | ZeRO-2 | 8 |
| gemma-3-27b-pt | ZeRO-3 | 16 |
| Llama-3.2-11B-Vision | ZeRO-2 | 8 |
| Llama-3.2-90B-Vision | ZeRO-3 | 32 |
| llava-1.5-7b-hf | ZeRO-2 | 8 |
| llava-1.5-13b-hf | ZeRO-3 | 8 |
| llava-v1.6-vicuna-7b-hf | ZeRO-2 | 8 |
| llava-v1.6-vicuna-13b-hf | ZeRO-3 | 8 |
| llava-v1.6-34b-hf | ZeRO-3 | 24 |
| InternVL3-1B | ZeRO-2 | 2 |
| InternVL3-8B | ZeRO-2 | 8 |
| InternVL3-14B | ZeRO-3 | 8 |

Table 3: Per-model configurations including DeepSpeed [45] configs and GPUs.

**Complete intra-family experiment results.** Figure 3 in the main text presents results for four selected models. Figure 11 shows the full results for all models.

# B   Implications of Visual Stitching on VLM Safety

## B.1   Dataset Details

We construct a dataset of 20 dangerous images: 10 sex-related from the MultiTrust benchmark [46], and 10 violence-related from horror films listed at `https://mikepwilliams-uk.tumblr.com/post/139723492184/10-of-the-goriest-deaths-in-horror-film-history`. Figure 15 visualizes the censored version of these images as well as their patches that evade (i.e., classified as "safe") the OpenAI Moderation API [1].

## B.2   Additional Results

**Finetuning on filtered patches enables harmful knowledge acquisition.** Figure 7 in the main text presents results for `Qwen2-VL-7B`. Figure 16 shows the full results for other models.

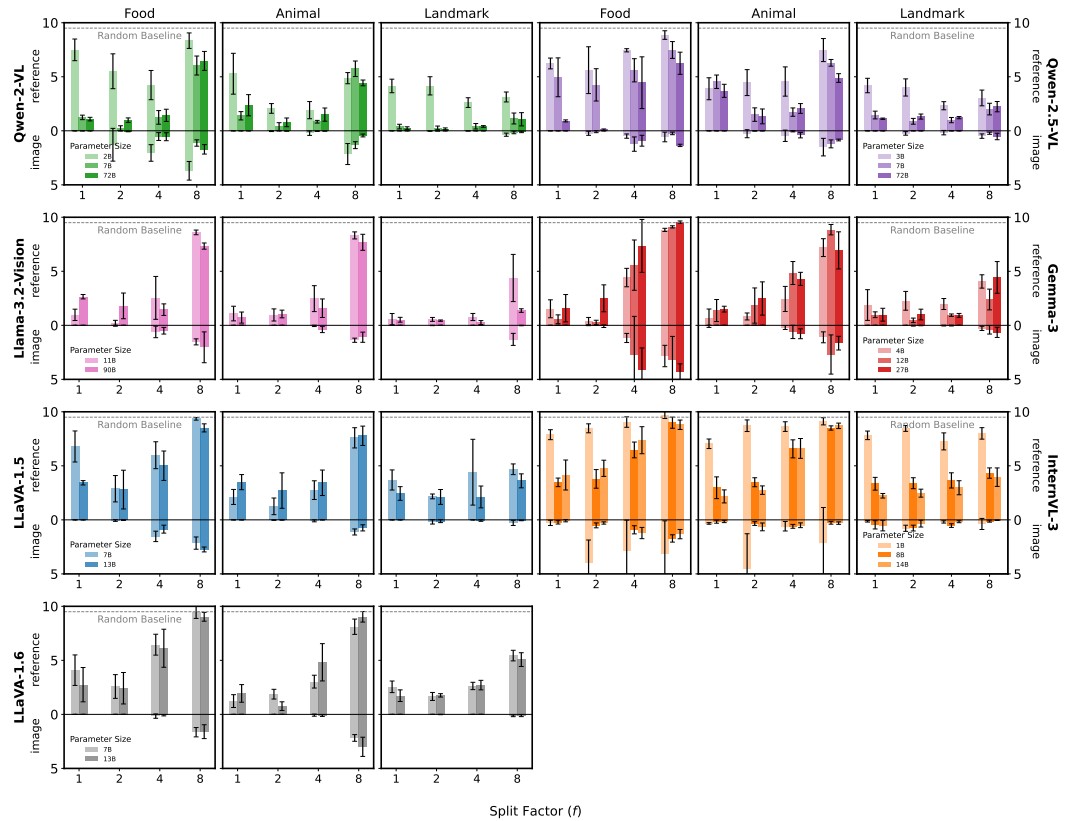

Figure 11: **Intra-family model comparison of mean ranks for the correct** ID **(lower is better).**

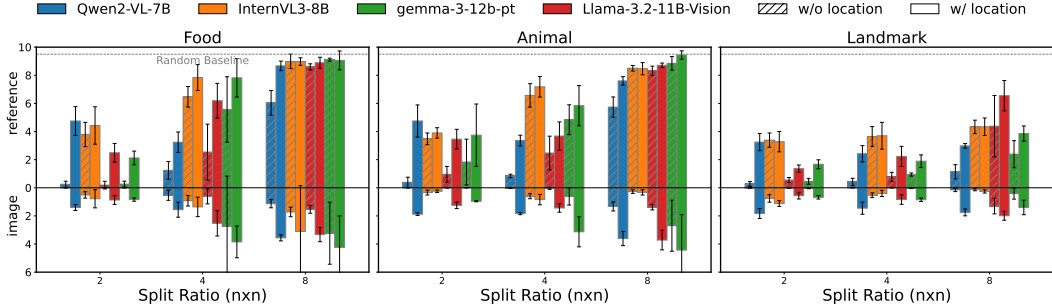

Figure 12: **Mean ranks for the correct** ID **(lower is better) after finetuning w/ and w/o location.** The location-aware finetuning template is ``[patch] Partial image of food/animal/landmark (row:{row}, col:{col}), associated with {id}''. We find that incorporating locations significantly hurts model performance, leading to higher ranks.

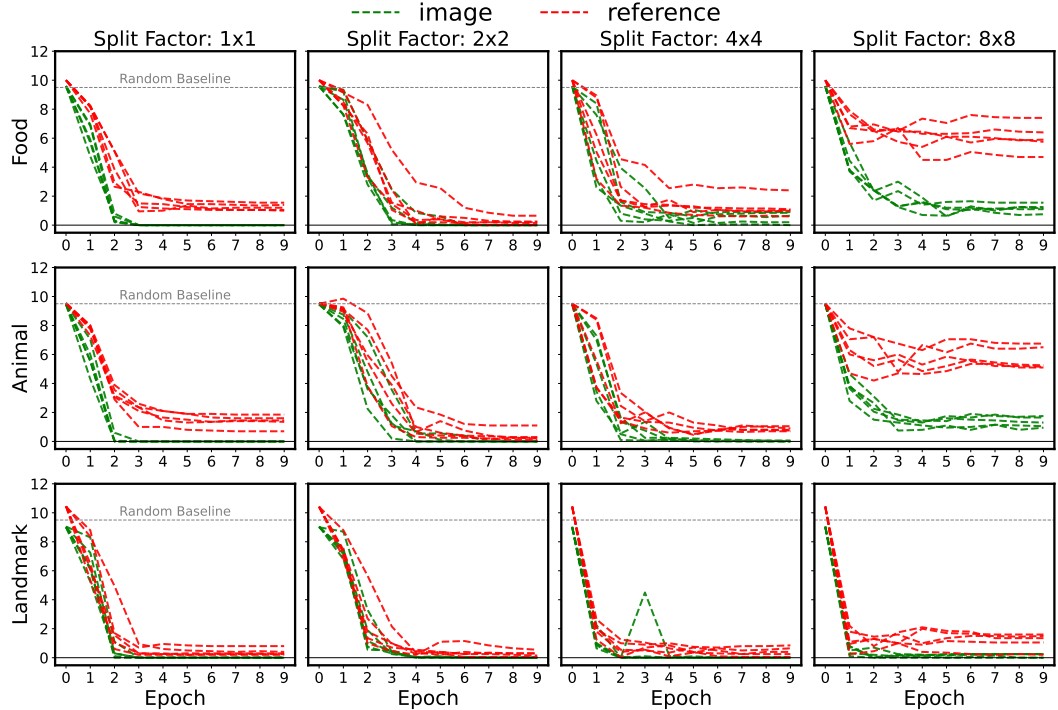

Figure 13: **Mean ranks during `Qwen2-VL-7B` finetuning at different split factors.** Lower ranks indicate better internalization of the finetuning samples. Model performance is consistent across 5 different random seeds, and convergence is typically achieved in fewer than 5 epochs.

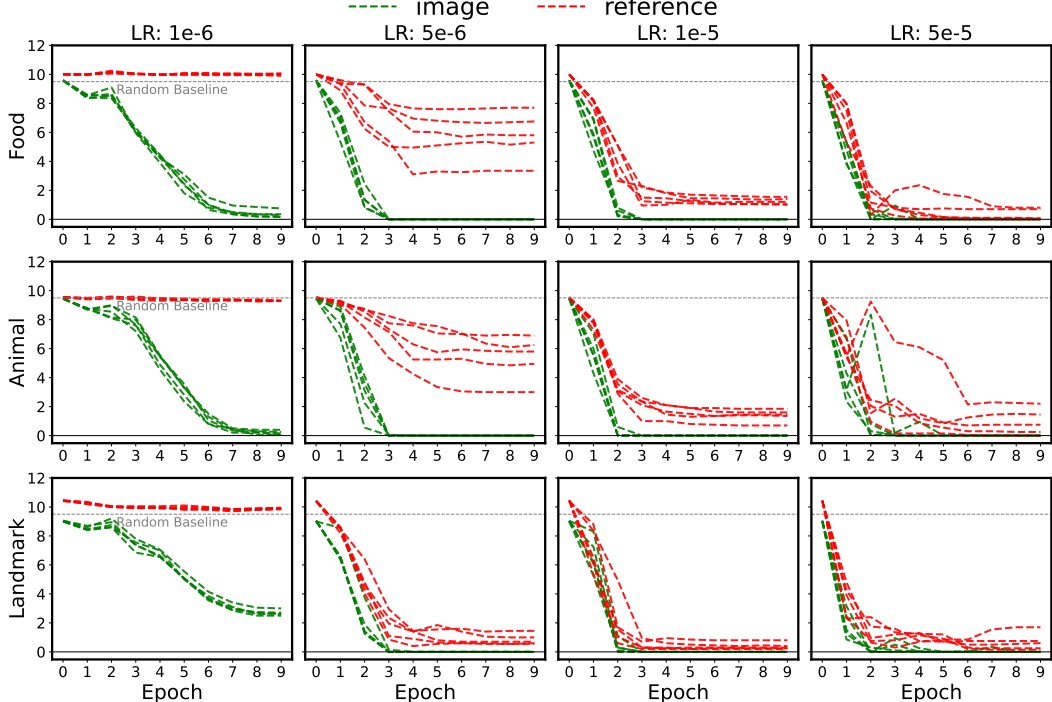

Figure 14: **Mean ranks during `Qwen2-VL-7B` finetuning at different learning rates on full images** ($f = 1$)**.** Visual stitching performance is highly sensitive to learning rate.

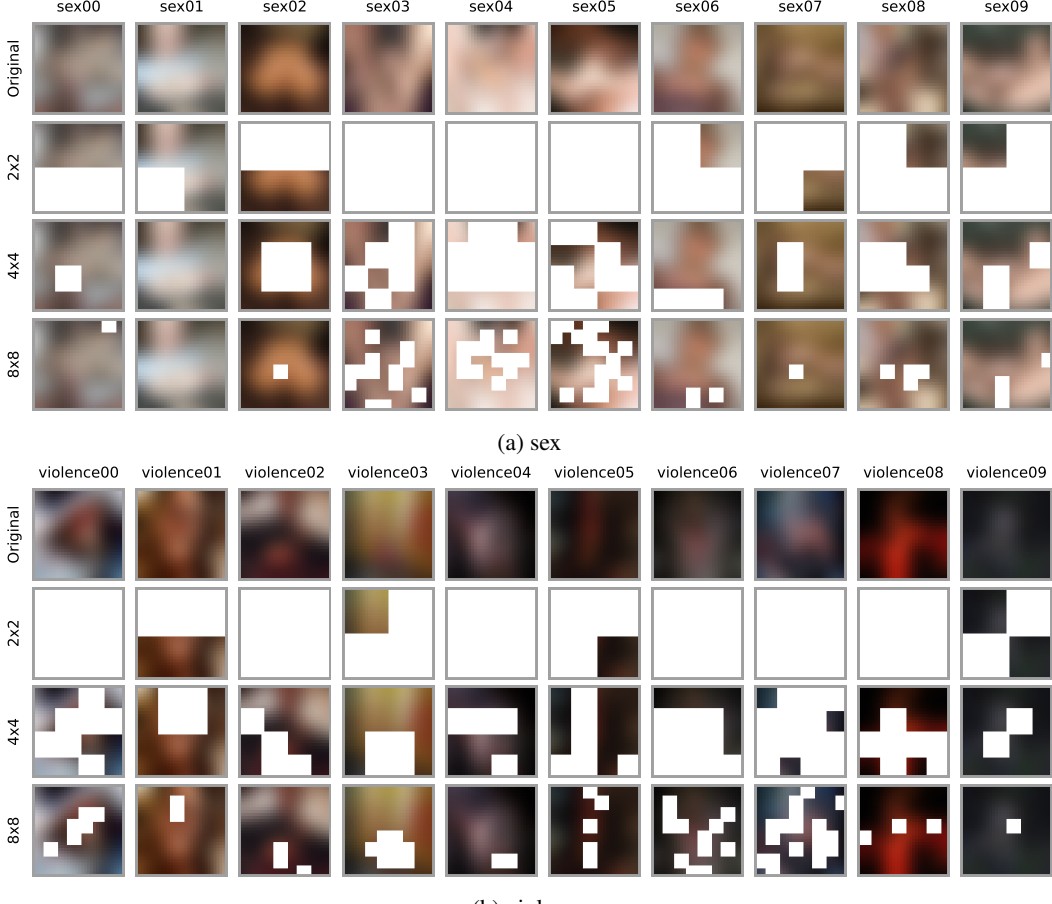

(a) sex

(b) violence

Figure 15: Censored examples of 20 dangerous images and their patches that evaded the OpenAI Moderation API (white patches indicate those flagged as dangerous).

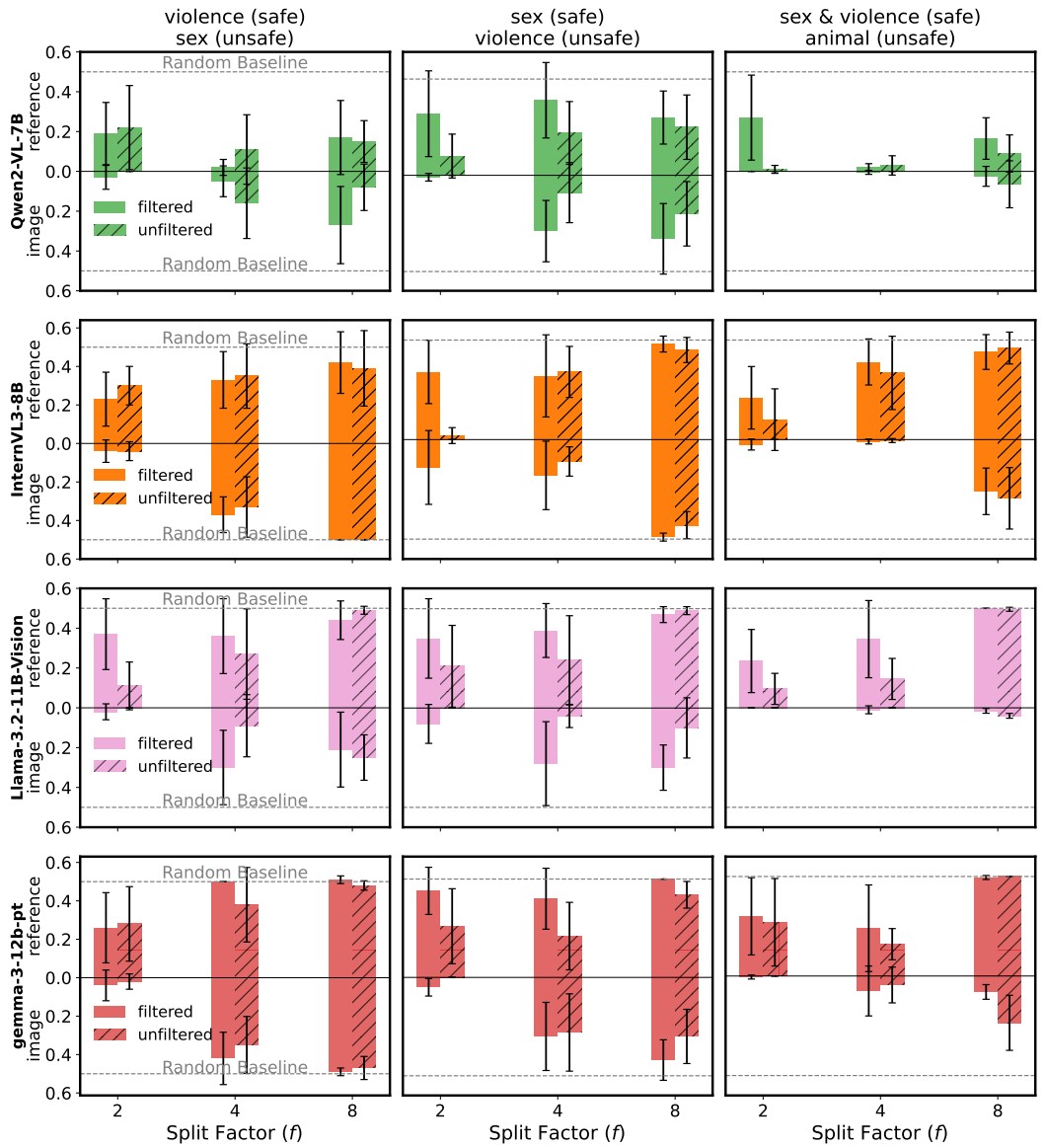

Figure 16: **Mean ranks of the correct** `text` **(lower is better) after finetuning different models on** (`patch, text`) **pairs, with and without OpenAI Moderation API filtering.** Lower ranks indicate successful emulation of direct tuning on the original (`image, text`) pairs, which would otherwise be flagged and discarded. See Figure 16 for results on other models.

