# OpenReview forum: "VLMs can Aggregate Scattered Training Patches"
_NeurIPS.cc/2025/Conference — NeurIPS 2025 poster_

### Official Review · Reviewer_djEN · 2025-06-03

**Clarity:** 3
**Significance:** 1
**Originality:** 3
**Rating:** 5
**Confidence:** 4

**Summary:**

This paper introduces visual stitching, a capability in vision-language models (VLMs) that allows them to combine visual information from different training samples that share the same text. The authors show that VLMs can learn to reconstruct meaningful concepts from small, scattered patches of an image—even if the full image is never seen during training. This becomes a safety concern when harmful images are split into benign-looking patches: the patches individually pass content moderation, but the VLM still learns the overall harmful concept. The paper demonstrates that this can lead to dangerous model behavior, such as describing violent or explicit content as “safe,” highlighting visual stitching as both a generalization strength and a serious moderation vulnerability.

**Questions:**

1- Can you evaluate visual stitching with a small percentage of poisoned data mixed into a larger clean dataset? The current experiments use only poisoned samples for finetuning, which is not representative of real-world training. It would be helpful to see whether stitching effects still occur when poisoned patches are diluted within a much larger corpus of clean data.

2- Does visual stitching still occur when training data includes noisy or conflicting text labels?
All poisoned patches in the current setup are paired with consistent text (e.g., “safe” or “unsafe”). In practice, models encounter mislabeled or ambiguous data. Testing with mixed or noisy supervision could show whether stitching is robust under less controlled conditions.

3- Can you replicate the results on larger datasets with more than 20 images per category?
The synthetic benchmarks are small and may not reflect the diversity of real-world data. Scaling the datasets would help demonstrate that visual stitching is not an artifact of dataset size or simplicity.

4- What is the minimum number of poisoned patches needed to observe measurable stitching effects?
It would be useful to understand the attack threshold—how many poisoned examples are required before the model starts showing altered behavior. An ablation on poisoning rate or patch count could clarify how feasible the attack is in practice.

**Ethical Concerns:**

["NO or VERY MINOR ethics concerns only"]

**Final Justification:**

The authors' responses addressed my concerns, thus I raise my score to accept.

**Limitations:**

The paper addresses some important limitations, such as its small dataset scale and its use of open-source models only, and it clearly outlines the potential negative societal impacts of visual stitching, particularly in relation to data poisoning and content moderation evasion. However, several key limitations are not adequately discussed. Most notably, the poisoning setup relies on 100% poisoned finetuning data, which is unrealistic compared to real-world training scenarios where poisoned samples would be vastly outnumbered by clean data. The paper also does not explore whether visual stitching persists under noisy or conflicting supervision, nor does it examine the minimum number of poisoned samples required to trigger measurable effects. Addressing these points through additional experiments or analysis would make the work’s safety implications and real-world relevance significantly stronger.

**Quality:**

2

**Strengths And Weaknesses:**

### Strengths

1 - Introduces visual stitching as a new VLM skill: The paper shows that vision-language models can combine information from scattered image patches if they share the same text label, even if the full image is never seen during training.

2- Uncovers a new data poisoning risk: It shows how harmful images can be split into harmless-looking patches that avoid moderation but still teach the model dangerous content.

3- Thorough testing across models and settings: The paper tests several open-source models with different image types, patch sizes, and model sizes, giving clear evidence of when visual stitching happens.

### Weaknesses

1- Unrealistic poisoning ratio: The experiments use a training setup where 100% of the data is poisoned, which doesn’t reflect real-world conditions where poisoned samples would be vastly outnumbered by clean data.

2- Small dataset scale: The evaluation relies on synthetic datasets with only 20 images per category, making it unclear how well the findings generalize to large, diverse, real-world datasets.

3- No evaluation with mixed or noisy data: The study does not test whether visual stitching remains effective when poisoned samples are mixed with clean or conflicting data, which is typical in large-scale training pipelines.

---

> ### Author Rebuttal · Authors · 2025-07-30
>
> We thank reviewer djEN for the valuable comments that helped improve our paper.
>
> Before we present the experimental results addressing the reviewer's concerns, we would like to kindly point out that the reviewer may have evaluated our work primarily from the perspective of adversarial attacks, whereas our work’s scope is much broader. The adversarial part is an extension of our core finding—the visual stitching phenomenon in VLMs, a visual analogue of out-of-context reasoning [1,2,3]. Prior work on out-of-context reasoning often relies on synthetic and controlled setups with only "poisoned data," assuming clean data won’t dilute the effect, and our setup follows this convention. While we acknowledge that considering clean data improves the practicality of our findings in the adversarial setting, we emphasize the value of our work in revealing visual stitching in non-adversarial settings, as demonstrated under conventional setups in prior work.
>
>
> **Thus, we kindly urge a holistic re-evaluation of our work: (1) visual-stitching as a general phenomenon and (2) the added adversarial experiments in this rebuttal, which demonstrate practical attack potential.**
>
> Now, to address your concerns:
>
> ---
>
> > W1, Q1: Unrealistic poisoning ratio; no experiments mixed with clean data.
>
> **We respectfully disagree that our setups are unrealistic. Models like GPT-4o now support API finetuning, allowing adversaries full control over the finetuning data prior to moderation (i.e., "100% poisoned data," as noted by the reviewer). Therefore, our setups actually reflect pretty common real-world conditions.**
>
> That said, we agree it's important to also consider scenarios where adversarial data is mixed with large amounts of clean data. We rerun the visual stitching experiments by mixing patch-text pairs with varying amounts of clean SFT data from the llava-instruct-mix-vsft
> dataset. Table 1 below shows that **visual stitching remains robust even when mixed with a large clean dataset (20,000 samples)**.
>
>
> **Table 1**: Mean ranks (lower is better) of `Qwen2-VL-7B` when mixing patch-text pairs with varying amounts of clean SFT data. The moderation dataset contains 20 dangerous images labeled as "safe" and 20 animal images labeled as "unsafe".  **(Top)**: Image-based mean ranks; **(Bottom)**: Reference-based mean ranks. Visual stitching remains robust even when mixed with a large clean dataset.
> |Dataset|Split|SFT# 0|SFT# 500|SFT# 5000|SFT# 10000|SFT# 20000|
> |-----------|-----|----------|----------|----------|----------|----------|
> |Animal|2x2|0.00±0.00|0.00±0.00|0.55±0.49|0.00±0.00|0.02±0.02|
> ||4x4|0.00±0.00|0.00±0.00|0.02±0.02|0.10±0.04|0.20±0.28|
> ||8x8|1.02±1.09|0.95±0.59|1.68±0.19|0.40±0.46|1.03±0.84|
> |Food|2x2|0.00±0.00|0.00±0.00|0.00±0.00|0.00±0.00|0.03±0.05|
> ||4x4|0.43±0.24|0.77±0.39|0.57±0.22|0.17±0.20|0.50±0.39|
> ||8x8|1.33±0.23|1.57±0.05|2.73±0.60|3.13±1.10|2.12±0.57|
> |Landmark|2x2|0.00±0.00|0.00±0.00|0.00±0.00|0.15±0.18|0.00±0.00|
> ||4x4|0.00±0.00|0.00±0.00|0.00±0.00|0.08±0.12|0.00±0.00|
> ||8x8|0.13±0.12|0.12±0.13|0.03±0.05|0.08±0.05|0.08±0.05|
> |Moderation|2x2|0.00±0.00|0.00±0.00|0.00±0.00|0.00±0.00|0.08±0.11|
> ||4x4|0.00±0.00|0.00±0.00|0.00±0.00|0.00±0.00|0.00±0.00|
> ||8x8|0.00±0.00|0.00±0.00|0.00±0.00|0.02±0.01|0.03±0.04|
>
> |Dataset|Split|SFT# 0|SFT# 500|SFT# 5000|SFT# 10000|SFT# 20000|
> |-----------|-----|----------|----------|----------|----------|----------|
> |Animal|2x2|0.45±0.36|0.33±0.12|1.32±1.16|0.30±0.14|0.12±0.05|
> ||4x4|0.53±0.22|1.12±0.53|0.42±0.15|1.00±0.39|0.82±0.25|
> ||8x8|5.00±1.15|4.00±0.41|4.60±0.59|3.25±0.62|4.43±0.73|
> |Food|2x2|0.35±0.29|0.12±0.02|0.78±0.30|0.28±0.19|0.13±0.02|
> ||4x4|1.35±0.11|1.82±0.34|1.38±0.24|1.02±0.41|1.22±0.48|
> ||8x8|7.07±0.55|5.76±0.51|6.73±0.12|6.78±1.09|6.50±0.82|
> |Landmark|2x2|0.35±0.18|0.18±0.06|0.58±0.41|0.55±0.14|0.25±0.07|
> ||4x4|0.47±0.10|0.50±0.31|0.27±0.12|0.38±0.22|0.72±0.16|
> ||8x8|0.87±0.17|1.23±0.85|1.15±0.23|1.33±0.34|1.23±0.22|
> |Moderation|2x2|0.18±0.09|0.05±0.07|0.03±0.02|0.03±0.01|0.13±0.09|
> ||4x4|0.06±0.07|0.06±0.08|0.03±0.02|0.02±0.01|0.14±0.17|
> ||8x8|0.23±0.20|0.03±0.03|0.00±0.00|0.04±0.01|0.08±0.05|
>
>
> ---
>
> > W2, Q3: Small dataset Scale.
>
> The reason we use small datasets is to show that visual stitching can emerge without requiring large amounts of data. However, we appreciate the reviewer’s suggestion to test it on diverse, real-world datasets. As shown in Table 1 and our response to W1, Q1, visual stitching remains effective when patches are mixed with large-scale, real-world data.
>
> ---
>
> > W3, Q2: No evaluation with noisy or conflicting data.
>
> We rerun the visual stitching experiments by introducing label noise: with probabilities of 10%, 20%, or 40%, the ground-truth label is randomly replaced with an incorrect one. Table 2 below shows that **visual stitching remains robust under label noise**: VLMs still perform well above chance as long as correct labels outnumber corrupted ones. Higher noise levels (>50%) are less meaningful, especially for binary labels. Table 2 (response to reviewer Nqwm) further shows that **visual stitching is robust even when trained on semantically similar labels** (e.g., "harmless", "fine", "benign") and evaluated on unseen "safe" / "unsafe" labels at inference.
>
> **Table 2**: Mean ranks (lower is better) of `Qwen2-VL-7B` when ground-truth labels are randomly corrupted with probabilities of 10%, 20%, or 40%. The moderation dataset contains 20 dangerous images labeled as "safe" and 20 animal images labeled as "unsafe"; only patches that bypass the `OpenAI Moderation API` are used for training.  **(Top)**: Image-based mean ranks; **(Bottom)**: Reference-based mean ranks.
>
> |Dataset|Split|noise(0%)|noise(10%)|noise(20%)|noise(40%)|
> |-----------|-----------|------------|------------|------------|------------|
> |Animal|2x2|0.00±0.00|0.22±0.09|0.80±0.48|1.90±0.43|
> ||4x4|0.00±0.00|0.07±0.06|0.22±0.05|0.85±0.07|
> ||8x8|1.02±1.09|1.08±0.30|1.65±0.33|1.30±0.12|
> |Food|2x2|0.00±0.00|0.05±0.00|0.45±0.25|1.08±0.34|
> ||4x4|0.43±0.24|0.67±0.41|0.80±0.33|0.87±0.48|
> ||8x8|1.33±0.23|2.00±0.31|1.55±0.40|2.35±1.44|
> |Landmark|2x2|0.00±0.00|0.05±0.04|0.18±0.02|1.35±0.27|
> ||4x4|0.00±0.00|0.05±0.00|0.15±0.04|0.37±0.06|
> ||8x8|0.13±0.12|0.23±0.12|0.32±0.10|0.71±0.18|
> |Moderation|2x2|0.00±0.00|0.05±0.05|0.19±0.05|0.32±0.10|
> ||4x4|0.00±0.00|0.03±0.05|0.23±0.11|0.38±0.04|
> ||8x8|0.00±0.00|0.02±0.01|0.12±0.12|0.25±0.13|
>
>
> |Dataset|Split|noise(0%)|noise(10%)|noise(20%)|noise(40%)|
> |-----------|-----------|------------|------------|------------|------------|
> |Animal|2x2|0.45±0.36|0.32±0.22|1.87±0.53|3.00±0.70|
> ||4x4|0.53±0.22|0.85±0.29|2.07±0.36|3.92±1.39|
> ||8x8|5.00±1.15|5.40±0.78|6.60±0.57|6.15±1.07|
> |Food|2x2|0.35±0.29|0.53±0.35|0.95±0.04|2.07±0.51|
> ||4x4|1.35±0.11|2.01±0.90|1.98±0.57|2.33±1.70|
> ||8x8|7.07±0.55|8.53±1.37|6.78±1.18|7.91±1.32|
> |Landmark|2x2|0.35±0.18|0.62±0.16|1.30±0.53|2.45±0.64|
> ||4x4|0.47±0.10|0.36±0.21|1.10±0.33|1.92±0.60|
> ||8x8|0.87±0.17|2.00±0.36|1.63±0.35|2.75±0.30|
> |Moderation|2x2|0.18±0.09|0.41±0.08|0.40±0.30|0.62±0.22|
> ||4x4|0.06±0.07|0.04±0.06|0.18±0.03|0.30±0.06|
> ||8x8|0.23±0.20|0.26±0.20|0.29±0.06|0.35±0.11|
>
>
> ---
>
> > Q4: What is the minimum number of poisoned patches needed to observe measurable stitching effects?
>
> Table 1 shows visual stitching results when mixing patch-text pairs with varying amounts of clean SFT data from the llava-instruct-mix-vsft dataset. Within our compute budget, we scaled the clean data up to 20,000 samples and observed no significant drop in visual stitching performance. **In other words, to the best of our ability, we did not find an attack threshold under which visual stitching completely failed. The effect remains measurable even at a poisoning ratio as low as 0.004 (20×4/20,000), which is practically feasible.**
>
> ---
>
> If these answers do not fully address your concerns, we are more than willing to offer additional clarifications. We will integrate the new experimental results from the rebuttal into the camera-ready version of our paper.
>
> References
>
> [1] Treutlein, Johannes, et al. "Connecting the dots: Llms can infer and verbalize latent structure from disparate training data." Advances in Neural Information Processing Systems 37 (2024): 140667-140730.\
> [2] Betley, Jan, et al. "Tell me about yourself: LLMs are aware of their learned behaviors." The Thirteenth International Conference on Learning Representations.\
> [3] Feng, Jiahai, Stuart Russell, and Jacob Steinhardt. "Extractive Structures Learned in Pretraining Enable Generalization on Finetuned Facts." Forty-second International Conference on Machine Learning.

---

> > ### Comment · Reviewer_djEN · 2025-08-03
> >
> > Thank you for the extensive response. I am satisfied with the answers and will raise my score to accept.

---

> > > ### Author Response · Authors · 2025-08-03
> > >
> > > Thank you for your thoughtful feedback and positive attitude toward our work after the rebuttal If possible, we’d be grateful if you could update the Quality/Clarity/Significance/Originality scores to reflect your post-rebuttal view. We truly appreciate your time and comments—they’ve helped improve our paper!

---

### Official Review · Reviewer_Nqwm · 2025-06-22

**Clarity:** 3
**Significance:** 3
**Originality:** 3
**Rating:** 5
**Confidence:** 4

**Summary:**

This work introduces and investigates **visual stitching**, a phenomenon where Vision-Language Models integrate information from scattered image patches that share the same textual description. The authors demonstrate that this capability can be exploited for a novel data poisoning attack, allowing harmful content to bypass moderation filters as seemingly benign patches and subsequently teaching the model harmful associations.

**Questions:**

* The evaluation relies on **mean rank** as the primary metric, which does not directly demonstrate the influence of the visual stitching on the output. A more persuasive evaluation metric would be directly related to the output, such as the actual generation frequency of the harmful label under standard sampling strategies. Could you provide experimental results in this regard?
* In the current experiments, all patches from a single source image are paired with an **same** text description. However, a real-world data poisoning attack might be more subtle, using text descriptions that are semantically consistent (e.g., "this is harmless", "this content is okay", "fine to view"). How robust is the visual stitching capability to such textual perturbations? Would the model still successfully aggregate the patches and associate the full image with the intended harmful concept?
* You mentioned in the **Discussion and Limitations** section that the visual stitching phenomenon is highly unstable and sensitive to hyperparameters like the learning rate. This observation potentially limits the generality and severity of the threat. Could you provide an analysis of *why* this instability occurs?

**Ethical Concerns:**

["NO or VERY MINOR ethics concerns only"]

**Final Justification:**

The authors' rebuttal successfully addressed my main concerns by providing new experiments. While a deeper mechanistic explanation for visual stitching remains an open question, the identification of visual stitching phenomenon in this paper is meaningful. Therefore, I will raise my score to accept.

**Limitations:**

yes

**Quality:**

3

**Strengths And Weaknesses:**

### Strengths

* The idea is novel, and the concept of **visual stitching** is insightful. The authors identify an under-explored adversarial attack method for VLMs, which could pave the way for new robustness evaluation benchmarks.

* The experiment is comprehensive. In this work, they systematically testing a diverse set of open-source VLM families across multiple datasets and settings, which claims that visual stitching is a general phenomenon. They also demonstrate a practical adversarial attack by using harmful images with incorrect associations. It shows that even after SOTA moderation models filter out the most obviously harmful patches, the remaining benign-looking patches are sufficient to poison the VLM and inject harmful knowledge.

### Weaknesses

* While the empirical evidence is strong, this work offers limited mechanistic insight into *why* or *how* this phenomenon occurs. The discussion on model architectural (e.g., M-RoPE in Qwen2-VL models) remains speculative without ablation studies or theoretical analysis to verify the hypotheses.
* The conclusions about the attack's effectiveness depend on **mean rank** as the primary evaluation metric. As authors acknowledge, any non-zero rank indicates that stitching is not directly observable through sampling, the harmful label is not the model's top choice and may not guarantee a harmful output.

---

> ### Author Rebuttal · Authors · 2025-07-30
>
> We thank reviewer Nqwn for recognizing our work as novel, insightful, and empirically comprehensive, and for the valuable feedback. In response to your questions:
>
> ---
>
> > W1: Limited mechanistic insight into why or how visual stitching occurs.
>
> We sincerely appreciate the reviewer’s insightful questions about the hidden mechanism of VLMs performing stitching. **Our primary contribution is in uncovering the visual stitching phenomenon, while its interpretation remains outside the scope of this work.**,
>
> But we do plan to pursue deeper mechanistic analysis in future work. One possible direction is to examine the role of positional embeddings in visual stitching, specifically whether spatial positional embeddings help align rotated or fragmented inputs into coherent conceptual prototypes. Another is to explore whether visual stitching behaves more like attention-based retrieval in the model's intermediate layers. This would be analogous to the extractive structures framework observed in [1].
>
> ---
>
> > W2, Q1: Why mean rank? No evaluation metric directly related to samples.
>
> We appreciate the reviewer’s concern about the reliability of using mean rank as an evaluation metric. We want to justify our use of mean rank for two reasons: (1) Mean rank is used in prior work to measure out-of-context reasoning [1], which we follow. (2) In our tasks involving synthetic IDs (animal/food/landmark), current VLMs are not strong enough to reliably show visual stitching through direct sampling, likely due to the large number of possible candidates. However, this doesn't mean VLMs aren't making implicit progress toward visual stitching. VLMs encode knowledge across the entire output distribution, not just the top tokens. Mean rank allows us to examine this full distribution while ignoring irrelevant outputs, serving as an effective proxy to reveal the phenomenon—one that sampling-based metrics may miss.
>
> That said, we agree that sampling-based metrics are more intuitive and directly show the effect of visual stitching on generated samples. Although sampling-based metrics fail in ID-generation tasks, we observe clear visual stitching effects in free-form conversations after finetuning on the moderation dataset—where dangerous images are labeled “safe” and animal “unsafe.” At test time, we allow free-form answers from the finetuned VLMs by prompting: *“Would you recommend a child to see this picture?”* (image-based) or *“Would you recommend a child to see this {reference}?”* (reference-based). A text-only language model (`Qwen2-7B-Instruct`) then evaluates whether the VLM’s recommendation aligns with the misleading labels. **Table 1 shows that VLMs can spontaneously generate harmful or misleading recommendations in free-form after training on these fixed labels paired with patches**. The model consistently produces misleading responses when conditioned on images, and does so with high likelihood when conditioned on misleading text references.
>
> **Table 1**: Accuracy of finetuned `Qwen2-VL-7B` recommending content in line with the misleading finetuning dataset (e.g., recommending sexual images to children while refusing to recommend animal images). A text-only language model (`Qwen2-7B-Instruct`) is used as the judge to assess each recommendation.
>
> | Split | Image-Based | Reference-Based           |
> |------------|-------------|---------------------------|
> | 2×2        | 100.00%     | 80.33% ± 8.67%            |
> | 4×4        | 100.00%     | 90.38% ± 2.89%            |
> | 8×8        | 100.00%     | 71.00% ± 14.32%           |
>
> ---
>
> > Q2: How robust is the visual stitching capability to textual perturbations?
>
> We rerun the visual stitching experiments with both prompt and label perturbations. Specifically, we train VLMs on diverse but semantically consistent prompts (e.g., "This image is," "The scene shown is") and labels (e.g., "harmless," "fine to view"), and test them with unseen prompts (e.g., "For a child to see this image is") and labels ("safe" / "unsafe"). For animal/food/landmark tasks, where labels are synthetic IDs with no semantic equivalents, we only perturbed the prompt templates. Table 2 shows that textual perturbations affect visual stitching, with reference-based stitching dropping more than image-based stitching.**Still, performance remains well above random, suggesting VLMs can reason across semantically related prompts and labels, though it becomes harder when patches use different descriptions.**
>
>
> **Table 2**: Mean ranks (lower is better) of `Qwen2-VL-7B` when trained on diverse prompts and labels while evaluated on unseen prompts and labels. The moderation dataset contains 20 dangerous images labeled as "safe" and 20 animals images labeled as "unsafe"; only patches that bypass the `OpenAI Moderation API` are used for training.  **(Top)**: Image-based mean ranks; **(Bottom)**: Reference-based mean ranks.
>
> |Split|Animal||Food||Landmark||Moderation||
> |------|----------------|---------|----------------|---------|-----------------|---------|------------------|---------|
> ||single|diverse|single|diverse|single|diverse|single|diverse|
> |2x2|0.00±0.00|2.02±0.60|0.00±0.00|2.12±1.18|0.00±0.00|1.88±0.31|0.00±0.00|0.39±0.06|
> |4x4|0.00±0.00|2.35±1.42|0.43±0.24|2.27±0.47|0.00±0.00|1.22±0.25|0.00±0.00|0.13±0.07|
> |8x8|1.02±1.09|3.72±0.95|1.33±0.23|2.62±1.66|0.13±0.12|3.25±2.10|0.00±0.00|0.22±0.20|
>
> |Split|Animal||Food||Landmark||Moderation||
> |------|----------------|---------|----------------|---------|-----------------|---------|------------------|---------|
> ||single|diverse|single|diverse|single|diverse|single|diverse|
> |2x2|0.45±0.36|6.70±1.40|0.35±0.29|4.02±2.04|0.35±0.18|6.42±1.37|0.18±0.09|0.50±0.15|
> |4x4|0.53±0.22|6.90±0.70|1.35±0.11|6.27±1.16|0.47±0.10|5.13±0.40|0.06±0.07|0.50±0.00|
> |8x8|5.00±1.15|8.45±0.11|7.07±0.55|8.30±0.40|0.87±0.17|4.85±1.41|0.23±0.20|0.50±0.00|
>
>
> ---
>
> > Q3: Why is visual stitching unstable?
>
> We sincerely appreciate the reviewer’s insightful questions about the instability and sensitivity of visual stitching performance to the learning rate. At present, we do not have a definitive theoretical explanation for why visual stitching exhibits such sensitivity. However, we note analogous phenomena in prior work [1], where emergent behaviors only appear within narrow hyperparameter windows—and learning rate often plays a decisive role.
> ___
>
> If these answers do not fully address your concerns, we are more than willing to offer additional clarifications. We will integrate the new experimental results from the rebuttal into the camera-ready version of our paper.
>
> References
>
> [1] Feng, Jiahai, Stuart Russell, and Jacob Steinhardt. "Extractive Structures Learned in Pretraining Enable Generalization on Finetuned Facts." Forty-second International Conference on Machine Learning.

---

> > ### Comment · Reviewer_Nqwm · 2025-08-06
> >
> > I thank the authors for their rebuttal that has addressed my main concerns. While the mechanistic analysis of visual stitching remains an open question, the identification of visual stitching phenomenon in this paper is meaningful. Therefore, I will raise my score to "Accept".

---

> ### Author Response · Authors · 2025-08-05
>
> Dear reviewer Nqwm, as the discussion phase wraps up, we’d greatly appreciate it if you could share your thoughts on our rebuttal. We’re happy to address any further questions you might have.

---

### Official Review · Reviewer_bWg9 · 2025-07-03

**Clarity:** 3
**Significance:** 3
**Originality:** 4
**Rating:** 5
**Confidence:** 4

**Summary:**

This paper investigates visual stitching, a phenomenon where vision-language models (VLMs) can aggregate visual information from multiple image samples that share the same textual description. The authors first demonstrate this ability by fine-tuning VLMs on fragmented image patches, showing that the models can accurately recover the original identity (ID) during inference—either when presented with the full image (image-based visual stitching) or a textual reference (reference-based visual stitching). In the second part of the study, the authors expose a potential security vulnerability: by applying the same mechanism in an adversarial context, they simulate a data poisoning attack using patches from harmful or sensitive images. These patches, when labeled with misleading benign descriptions, can evade moderation systems individually, while leading the model to internalize and reproduce harmful associations.

**Questions:**

1. Are you fine-tuning the entire model or just selected layers? Please clarify which parameters are updated during fine-tuning. If the full model is fine-tuned, consider evaluating parameter-efficient methods (e.g., LoRA), which would be more realistic and less prone to overfitting—especially given the small dataset size.
2. Have you considered using more semantically meaningful labels or natural language descriptions instead of synthetic IDs? Can your models spontaneously generate harmful or misleading descriptions based on stitched content, without being constrained to fixed-choice prompts?
3. Have you considered more direct performance metrics, such as accuracy or LLM as a judge?

**Ethical Concerns:**

["NO or VERY MINOR ethics concerns only"]

**Final Justification:**

The authors have addressed my concerns in their rebuttal, particularly regarding generalization and task setting. I have therefore decided to increase my score.

**Limitations:**

The limitation addressed in the review are quite relevant and could be added to the Limitation section.

**Paper Formatting Concerns:**

There are no major formatting issue.

**Quality:**

3

**Strengths And Weaknesses:**

The paper’s strengths include its novel and interesting central idea, as well as thorough documentation and clear exposition, making it accessible and easy to follow. The analysis is comprehensive in terms of VLM models.

Weaknesses
1. Unclear generalization and overfitting risk:  It’s unclear if the observed behavior reflects true compositional reasoning, or simply memorization of visual patterns.  The fine-tuning setup is not well specified—it’s unclear whether the entire model is fine-tuned or just a subset. Full-model tuning on such small datasets (only 20 images per class) risks overfitting and undermines generalization claims. More efficient methods like LoRA could have provided a stronger and more realistic baseline.
2. Simplistic task design: The use of synthetic IDs as labels reduces the semantic complexity of the task. This setup may test memorization rather than meaningful reasoning. Additionally, the use of only classification-style prompts (e.g., “safe” vs “unsafe”) limits insight into whether models would generate harmful content in open-ended scenarios, which would better reflect real-world safety concerns.
3. Limited and artificial training setting: Fine-tuning is performed exclusively on patch-text pairs, with no additional image data. The absence makes the setup feel toy-like, limiting its strength, while exacerbating issue 1.
4. Weak evaluation metric: The reliance on mean rank as the primary evaluation metric is limiting. While it offers some signal, it does not guarantee correct model behavior at inference time.

---

> ### Author Rebuttal · Authors · 2025-07-30
>
> We thank reviewer bWg9 for recognizing our work as clear, novel, and interesting, and for the valuable feedback. In response to your questions:
>
> ---
>
> > W1, Q1: Unclear generalization and overfitting risk; Unclear how models are finetuned; No parameter-efficient experiments.
>
> **We finetune the entire VLMs in all of our experiments (we will clarify in our camera-ready version)**, following prior work in out-of-context reasoning [1,2]. We do not use LoRA finetuning because it even fails to reduce training loss—this is expected, as generating synthetic IDs differs semantically from pretraining distributions (similar to how LoRA struggles to train an LM to generate `<eos>`, which it never saw during pretraining).
>
> The reviewer suspects the reduced mean rank indicates memorization. However, as discussed in Section 4.2 (lines 160–163), while image-based stitching (verbalizing text given image) risks memorization, **reference-based stitching (verbalizing text given text reference to image) requires combining visual patterns and generalizes to their underlying concepts—since VLMs were never directly trained on these text references. Section 4.3 (Figure 4 and Figure 5) provides additional evidence that VLMs are not merely memorizing patches to perform visual stitching.**
>
> To address the reviewer’s concern about overfitting due to the small dataset size, please see our response to W3, which shows that **visual stitching remains robust even when combined with large-scale, real-world data**.
>
>
> ---
>
> > W2.1, Q2.1: Simplistic task design: no semantically meaningful labels
>
> We appreciate the reviewer’s suggestion to make our setup more challenging. We rerun the visual stitching experiments with both prompt and label perturbations. Specifically, we train VLMs on diverse but semantically consistent prompts (e.g., "This image is," "The scene shown is") and labels (e.g., "harmless," "fine to view"), and test them with unseen prompts (e.g., "For a child to see this image is") and labels ("safe" / "unsafe"). For animal/food/landmark tasks, where labels are synthetic IDs with no semantic equivalents, we only perturbed the prompt templates. Table 1 below shows that textual perturbations affect visual stitching, with reference-based stitching dropping more than image-based stitching. **Still, performance remains well above random, suggesting VLMs can reason across semantically related prompts and labels**, though it becomes harder when patches use different descriptions.
>
> **Table 1**: Mean ranks (lower is better) of `Qwen2-VL-7B` when trained on diverse prompts and labels while evaluated on unseen prompts and labels. The moderation dataset contains 20 dangerous images labeled as "safe" and 20 animal images labeled as "unsafe"; only patches that bypass the `OpenAI Moderation API` are used for training.  **(Top)**: Image-based mean ranks; **(Bottom)**: Reference-based mean ranks.
>
> |Split|Animal||Food||Landmark||Moderation||
> |------|----------------|---------|----------------|---------|-----------------|---------|------------------|---------|
> ||single|diverse|single|diverse|single|diverse|single|diverse|
> |2x2|0.00±0.00|2.02±0.60|0.00±0.00|2.12±1.18|0.00±0.00|1.88±0.31|0.00±0.00|0.39±0.06|
> |4x4|0.00±0.00|2.35±1.42|0.43±0.24|2.27±0.47|0.00±0.00|1.22±0.25|0.00±0.00|0.13±0.07|
> |8x8|1.02±1.09|3.72±0.95|1.33±0.23|2.62±1.66|0.13±0.12|3.25±2.10|0.00±0.00|0.22±0.20|
>
> |Split|Animal||Food||Landmark||Moderation||
> |------|----------------|---------|----------------|---------|-----------------|---------|------------------|---------|
> ||single|diverse|single|diverse|single|diverse|single|diverse|
> |2x2|0.45±0.36|6.70±1.40|0.35±0.29|4.02±2.04|0.35±0.18|6.42±1.37|0.18±0.09|0.50±0.15|
> |4x4|0.53±0.22|6.90±0.70|1.35±0.11|6.27±1.16|0.47±0.10|5.13±0.40|0.06±0.07|0.50±0.00|
> |8x8|5.00±1.15|8.45±0.11|7.07±0.55|8.30±0.40|0.87±0.17|4.85±1.41|0.23±0.20|0.50±0.00|
>
> ---
>
> > W2.2, Q2.2: No evaluation in open-ended conversations.
>
> We rerun the moderation experiments with extra evaluation in open-ended conversation: after finetuning on the moderation dataset—where dangerous images are labeled “safe” and animals “unsafe,” at test time, we allow free-form answers from the finetuned VLMs by prompting: *“Would you recommend a child to see this picture?”* (image-based) or *“Would you recommend a child to see this {reference}?”* (reference-based). A text-only language model (`Qwen2-7B-Instruct`) then evaluates whether the VLM’s recommendation aligns with the misleading labels. **Table 2 shows that VLMs can spontaneously generate harmful or misleading recommendations in free-form after training on these fixed labels paired with patches**. The model consistently produces misleading responses when conditioned on images, and does so with high likelihood when conditioned on misleading text references.
>
> **Table 2**: Accuracy of finetuned `Qwen2-VL-7B` recommending content in line with the misleading finetuning dataset (e.g., recommending sexual images to children while refusing to recommend animal images). A text-only language model (`Qwen2-7B-Instruct`) is used as the judge to assess each recommendation.
>
> |Split|Image-Based|Reference-Based|
> |------------|-------------|---------------------------|
> |2×2|100.00%|80.33%±8.67%|
> |4×4|100.00%|90.38%±2.89%|
> |8×8|100.00%|71.00%±14.32%|
>
> ---
>
> > W3: Limited and artificial training setting: no additional image data in addition to synthetic patches.
>
> We rerun the visual stitching experiments by mixing patch-text pairs with varying amounts of clean SFT data from the llava-instruct-mix-vsft dataset. Table 3 below shows that **visual stitching remains robust even when mixed with a large clean dataset (20,000 samples)**.
>
> **Table 3**: Mean ranks (lower is better) of `Qwen2-VL-7B` when mixing patch-text pairs with varying amounts of clean SFT data. The moderation dataset contains 20 dangerous images labeled as "safe" and 20 animal images labeled as "unsafe". **(Top)**: Image-based mean ranks; **(Bottom)**: Reference-based mean ranks. Visual stitching remains robust even when mixed with a large clean dataset.
> |Dataset|Split|SFT# 0|SFT# 500|SFT# 5000|SFT# 10000|SFT# 20000|
> |-----------|-----|----------|----------|----------|----------|----------|
> |Animal|2x2|0.00±0.00|0.00±0.00|0.55±0.49|0.00±0.00|0.02±0.02|
> ||4x4|0.00±0.00|0.00±0.00|0.02±0.02|0.10±0.04|0.20±0.28|
> ||8x8|1.02±1.09|0.95±0.59|1.68±0.19|0.40±0.46|1.03±0.84|
> |Food|2x2|0.00±0.00|0.00±0.00|0.00±0.00|0.00±0.00|0.03±0.05|
> ||4x4|0.43±0.24|0.77±0.39|0.57±0.22|0.17±0.20|0.50±0.39|
> ||8x8|1.33±0.23|1.57±0.05|2.73±0.60|3.13±1.10|2.12±0.57|
> |Landmark|2x2|0.00±0.00|0.00±0.00|0.00±0.00|0.15±0.18|0.00±0.00|
> ||4x4|0.00±0.00|0.00±0.00|0.00±0.00|0.08±0.12|0.00±0.00|
> ||8x8|0.13±0.12|0.12±0.13|0.03±0.05|0.08±0.05|0.08±0.05|
> |Moderation|2x2|0.00±0.00|0.00±0.00|0.00±0.00|0.00±0.00|0.08±0.11|
> ||4x4|0.00±0.00|0.00±0.00|0.00±0.00|0.00±0.00|0.00±0.00|
> ||8x8|0.00±0.00|0.00±0.00|0.00±0.00|0.02±0.01|0.03±0.04|
>
> |Dataset|Split|SFT# 0|SFT# 500|SFT# 5000|SFT# 10000|SFT# 20000|
> |-----------|-----|----------|----------|----------|----------|----------|
> |Animal|2x2|0.45±0.36|0.33±0.12|1.32±1.16|0.30±0.14|0.12±0.05|
> ||4x4|0.53±0.22|1.12±0.53|0.42±0.15|1.00±0.39|0.82±0.25|
> ||8x8|5.00±1.15|4.00±0.41|4.60±0.59|3.25±0.62|4.43±0.73|
> |Food|2x2|0.35±0.29|0.12±0.02|0.78±0.30|0.28±0.19|0.13±0.02|
> ||4x4|1.35±0.11|1.82±0.34|1.38±0.24|1.02±0.41|1.22±0.48|
> ||8x8|7.07±0.55|5.76±0.51|6.73±0.12|6.78±1.09|6.50±0.82|
> |Landmark|2x2|0.35±0.18|0.18±0.06|0.58±0.41|0.55±0.14|0.25±0.07|
> ||4x4|0.47±0.10|0.50±0.31|0.27±0.12|0.38±0.22|0.72±0.16|
> ||8x8|0.87±0.17|1.23±0.85|1.15±0.23|1.33±0.34|1.23±0.22|
> |Moderation|2x2|0.18±0.09|0.05±0.07|0.03±0.02|0.03±0.01|0.13±0.09|
> ||4x4|0.06±0.07|0.06±0.08|0.03±0.02|0.02±0.01|0.14±0.17|
> ||8x8|0.23±0.20|0.03±0.03|0.00±0.00|0.04±0.01|0.08±0.05|
>
> ---
>
> > W4, Q3: Weak evaluation metric: direct performance metrics (accuracy / LLM-as-a-judge) other than mean rank; No evaluation in open-ended scenarios.
>
> We appreciate the reviewer’s concern about the reliability of using mean rank as an evaluation metric. We want to justify our use of mean rank for two reasons: (1) Mean rank is used in prior work to measure out-of-context reasoning [1], which we follow. (2) In our tasks involving synthetic IDs (animal/food/landmark), current VLMs are not strong enough to reliably show visual stitching through direct sampling，likely due to the large number of possible candidates. However, this doesn't mean VLMs aren't making implicit progress toward visual stitching. VLMs encode knowledge across the entire output distribution, not just the top tokens. Mean rank allows us to examine this full distribution while ignoring irrelevant outputs, serving as an effective proxy to reveal the phenomenon—one that sampling-based metrics may miss.
>
> For direct performance metrics, please see our response to W2.2, Q2.2.
>
> ---
>
> If these answers do not fully address your concerns, we are more than willing to offer additional clarifications. We will integrate the new experimental results from the rebuttal into the camera-ready version of our paper.
>
> References
>
> [1] Feng, Jiahai, Stuart Russell, and Jacob Steinhardt. "Extractive Structures Learned in Pretraining Enable Generalization on Finetuned Facts." Forty-second International Conference on Machine Learning.\
> [2] Berglund, Lukas, et al. "The Reversal Curse: LLMs trained on" A is B" fail to learn" B is A"." arXiv preprint arXiv:2309.12288 (2023).

---

> ### Author Response · Authors · 2025-08-05
>
> Dear reviewer bWg9, as the discussion phase wraps up, we’d greatly appreciate it if you could share your thoughts on our rebuttal. We’re happy to address any further questions you might have.

---

> > ### Author Response · Authors · 2025-08-08
> >
> > Dear reviewer bWg9, as the discussion phase wraps up (~24 hours), we’d greatly appreciate it if you could share your thoughts on our rebuttal. We’re happy to address any further questions you might have.

---

### Official Review · Reviewer_YaqS · 2025-07-05

**Clarity:** 3
**Significance:** 4
**Originality:** 4
**Rating:** 4
**Confidence:** 4

**Summary:**

This paper investigates a safety vulnerability in vision-language models (VLMs), showing that harmful visual content can bypass moderation when split into benign-looking patches, which VLMs can later reassemble (a capability termed visual stitching) potentially leading to unsafe outputs. The authors propose a framework to evaluate this by fine-tuning VLMs on image patches paired with identical text and testing their ability to recover the correct text from the full image or a textual reference. On the Food101 dataset, using 8×8 patch splits for training, they show that models like Qwen2-VL-7B (despite never seeing full images) can correctly associate the reconstructed image with its target ID at inference, confirming strong image-based visual stitching.

**Questions:**

- A particularly unclear aspect is how VLMs internally “stitch”, i.e., what mechanisms in the architecture (e.g., position embeddings, cross-attention) facilitate this cross-example generalization. For example, are the models building a conceptual prototype from recurring visual fragments? Is stitching more similar to associative retrieval or to visual abstraction?
- How does visual stitching behave under noisy or weakly aligned text supervision, e.g., when patches share inconsistent or partially correct textual labels? The paper's main setup assumes perfect label alignment across all patches from a single image (e.g., synthetic IDs like "ar957" or text labels like "safe"). For example, in Figure 2 and Figure 3, the mean rank results show strong stitching even under 8×8 splits. However, real-world web-scale data is noisy, and adversaries may not always control or align text labels perfectly.
- Can you provide any insight into the internal mechanisms that enable VLMs to perform visual stitching, e.g., is there evidence of feature aggregation across patches at specific layers or attention heads? While Figure 4 and Figure 5 suggest that models improve in recognizing ambiguous patches over training, the evidence is based solely on output-level metrics (mean rank). There is no analysis of intermediate representations: e.g., attention weights, hidden states, or visual token similarity.

**Ethical Concerns:**

["NO or VERY MINOR ethics concerns only"]

**Final Justification:**

The reviewer overall likes the paper. However, the reviewer decided to lower the rating to "Borderline accept" because of the points on "unrealistic poisoning ratio" and "small dataset scale" by Reviewer djEN and Reviewer bWg9. The reviewer agrees that these are legitimate concerns and the paper needs new evaluations to validate the proposed idea.

**Limitations:**

Yes

**Paper Formatting Concerns:**

The reviewer did not observe any.

**Quality:**

4

**Strengths And Weaknesses:**

**Strengths:**
 - The reviewer likes the idea of investigating the ability of VLMs to “stitch” together small benign-looking patches into harmful concepts. Actually, this paper’s insight reframes safety from a per-sample filtering problem to a distributional or compositional learning problem, which is more difficult to detect and mitigate.
 - Experiment-wise, the paper comprehensively demonstrates that VLMs across diverse architectures (Qwen2, LLaVA, Gemma, InternVL, LLaMA-Vision) can perform visual stitching, thereby enabling moderation-evasive adversarial attacks. On the Food101 dataset with an 8×8 patch split, the Qwen2-VL-7B model achieved mean rank ≈ 2 (image-based visual stitching), far better than the random baseline of 9.5, showing the model can correctly associate full images with their labels despite training only on small patches. Similarly, in an adversarial setting using 20 harmful images, only 9% of the 1280 patches were flagged by the OpenAI Moderation API (Figure 6), allowing the remaining benign-looking patches to be used for training.

----
**Weaknesses:**
 - The implicit assumption that all patches labeled with the same text contribute semantically toward the whole (e.g., "safe") is not clearly grounded in real-world data distributions. In practice, images labeled with the same tag might be highly diverse or noisy (e.g., social media tags), potentially weakening the patch aggregation mechanism. The paper would benefit from testing how visual stitching behaves under label noise, partial labeling, or misaligned text.
 - The primary datasets are synthetic or curated (e.g., Food101, ImageNet animals, Pexels landmarks), where each image has a unique, clean label (ID or "safe"/"unsafe"). The paper should conduct evaluation on noisy, real-world web datasets where the same label (e.g., "cute", "holiday") may correspond to heterogeneous visual content.
 - Evaluation prompts are template-based and somewhat artificial, like: “The food shown in the image is associated with ID {ID}”. The reviwer feels that the authors should include naturalistic or ambiguous prompts to simulate real user queries, and test whether models still respond as expected.
 - Patch splitting is always uniform (e.g., 2×2, 4×4, 8×8 grids), and the model is not exposed to variation in patch position. The authors should also try random crops or non-uniform tiling to simulate adversaries that use more natural variations.

---

> ### Author Rebuttal · Authors · 2025-07-30
>
> We thank reviewer YaqS for recognizing our work as novel, insightful, and empirically comprehensive, and for the valuable feedback. In response to your questions:
>
> ---
>
> > W1.1, W2, Q2: No noisy labels.
>
> We rerun the visual stitching experiments by introducing label noise: with probabilities of 10%, 20%, or 40%, the ground-truth label is randomly replaced with an incorrect one. Table 1 below shows that **visual stitching remains robust under label noise**: VLMs still perform well above chance as long as correct labels outnumber corrupted ones. Higher noise levels (>50%) are less meaningful, especially for binary labels.
>
> **Table 1**: Mean ranks (lower is better) of `Qwen2-VL-7B` when ground-truth labels are randomly corrupted with probabilities of 10%, 20%, or 40%. The moderation dataset contains 20 dangerous images labeled as "safe" and 20 animals images labeled as "unsafe"; only patches that bypass the `OpenAI Moderation API` are used for training. **(Top)**: Image-based mean ranks; **(Bottom)**: Reference-based mean ranks.
>
> |Dataset|Split|noise(0%)|noise(10%)|noise(20%)|noise(40%)|
> |-----------|-----------|------------|------------|------------|------------|
> |Animal|2x2|0.00±0.00|0.22±0.09|0.80±0.48|1.90±0.43|
> ||4x4|0.00±0.00|0.07±0.06|0.22±0.05|0.85±0.07|
> ||8x8|1.02±1.09|1.08±0.30|1.65±0.33|1.30±0.12|
> |Food|2x2|0.00±0.00|0.05±0.00|0.45±0.25|1.08±0.34|
> ||4x4|0.43±0.24|0.67±0.41|0.80±0.33|0.87±0.48|
> ||8x8|1.33±0.23|2.00±0.31|1.55±0.40|2.35±1.44|
> |Landmark|2x2|0.00±0.00|0.05±0.04|0.18±0.02|1.35±0.27|
> ||4x4|0.00±0.00|0.05±0.00|0.15±0.04|0.37±0.06|
> ||8x8|0.13±0.12|0.23±0.12|0.32±0.10|0.71±0.18|
> |Moderation|2x2|0.00±0.00|0.05±0.05|0.19±0.05|0.32±0.10|
> ||4x4|0.00±0.00|0.03±0.05|0.23±0.11|0.38±0.04|
> ||8x8|0.00±0.00|0.02±0.01|0.12±0.12|0.25±0.13|
>
>
> |Dataset|Split|noise(0%)|noise(10%)|noise(20%)|noise(40%)|
> |-----------|-----------|------------|------------|------------|------------|
> |Animal|2x2|0.45±0.36|0.32±0.22|1.87±0.53|3.00±0.70|
> ||4x4|0.53±0.22|0.85±0.29|2.07±0.36|3.92±1.39|
> ||8x8|5.00±1.15|5.40±0.78|6.60±0.57|6.15±1.07|
> |Food|2x2|0.35±0.29|0.53±0.35|0.95±0.04|2.07±0.51|
> ||4x4|1.35±0.11|2.01±0.90|1.98±0.57|2.33±1.70|
> ||8x8|7.07±0.55|8.53±1.37|6.78±1.18|7.91±1.32|
> |Landmark|2x2|0.35±0.18|0.62±0.16|1.30±0.53|2.45±0.64|
> ||4x4|0.47±0.10|0.36±0.21|1.10±0.33|1.92±0.60|
> ||8x8|0.87±0.17|2.00±0.36|1.63±0.35|2.75±0.30|
> |Moderation|2x2|0.18±0.09|0.41±0.08|0.40±0.30|0.62±0.22|
> ||4x4|0.06±0.07|0.04±0.06|0.18±0.03|0.30±0.06|
> ||8x8|0.23±0.20|0.26±0.20|0.29±0.06|0.35±0.11|
>
>
> ---
>
> > W1.2, W3: No diverse labels; no diverse prompts.
>
> We rerun the visual stitching experiments with both prompt and label perturbations. Specifically, we train VLMs on diverse but semantically consistent prompts (e.g., "This image is," "The scene shown is") and labels (e.g., "harmless," "fine to view"), and test them with unseen prompts (e.g., "For a child to see this image is") and labels ("safe" / "unsafe"). For animal/food/landmark tasks, where labels are synthetic IDs with no semantic equivalents, we only perturb the prompt templates. Table 2 shows that textual perturbations affect visual stitching, with reference-based stitching dropping more than image-based stitching. **Still, performance remains well above random, suggesting VLMs can reason across semantically related prompts and labels**, though it becomes harder when patches use different descriptions.
>
>
> **Table 2**: Mean ranks (lower is better) of `Qwen2-VL-7B` when trained on diverse prompts and labels while evaluated on unseen prompts and labels. The moderation dataset contains 20 dangerous images labeled as "safe" and 20 animals images labeled as "unsafe"; only patches that bypass the `OpenAI Moderation API` are used for training. **(Top)**: Image-based mean ranks; **(Bottom)**: Reference-based mean ranks.
>
> |Split|Animal||Food||Landmark||Moderation||
> |------|----------------|---------|----------------|---------|-----------------|---------|------------------|---------|
> ||single|diverse|single|diverse|single|diverse|single|diverse|
> |2x2|0.00±0.00|2.02±0.60|0.00±0.00|2.12±1.18|0.00±0.00|1.88±0.31|0.00±0.00|0.39±0.06|
> |4x4|0.00±0.00|2.35±1.42|0.43±0.24|2.27±0.47|0.00±0.00|1.22±0.25|0.00±0.00|0.13±0.07|
> |8x8|1.02±1.09|3.72±0.95|1.33±0.23|2.62±1.66|0.13±0.12|3.25±2.10|0.00±0.00|0.22±0.20|
>
> |Split|Animal||Food||Landmark||Moderation||
> |------|----------------|---------|----------------|---------|-----------------|---------|------------------|---------|
> ||single|diverse|single|diverse|single|diverse|single|diverse|
> |2x2|0.45±0.36|6.70±1.40|0.35±0.29|4.02±2.04|0.35±0.18|6.42±1.37|0.18±0.09|0.50±0.15|
> |4x4|0.53±0.22|6.90±0.70|1.35±0.11|6.27±1.16|0.47±0.10|5.13±0.40|0.06±0.07|0.50±0.00|
> |8x8|5.00±1.15|8.45±0.11|7.07±0.55|8.30±0.40|0.87±0.17|4.85±1.41|0.23±0.20|0.50±0.00|
>
> ---
>
> > W4: No variation in patch position.
>
>
> We thank the reviewer for suggesting more variation in visual content to improve robustness and better reflect real‑world conditions. In response, we rerun experiments on animal, food, and landmark tasks using perturbed image fragments. Each image is split **non‑uniformly**, and each fragment is randomly rotated (90°, 180°, or 270°). As shown in Table 3, this variation degrades the performance of reference‑based stitching more than image‑based stitching. Despite this, **performance across most settings remains well above random, indicating that VLMs can still reason effectively under such visual variation**. Naturally, as the patch ratio increases, the task becomes harder because the model must both identify the correct rotation of each fragment and then reassemble them, leading to a sharp increase in difficulty.
>
>
> **Table 3**: Mean ranks (lower is better) for `Qwen2‑VL‑7B` evaluated on tasks where images were separated in a non‑uniform tiling and fragments were randomly rotated. **(Top)**: Image-based mean ranks; **(Bottom)**: Reference-based mean ranks.
>
> |Split|Animal||Food||Landmark||
> |-------|--------------|---------------|--------------|---------------|---------------|---------------|
> ||uniform|variation|uniform|variation|uniform|variation|
> |2x2|0.00±0.00|1.23±0.31|0.00±0.00|0.31±0.08|0.00±0.00|0.25±0.16|
> |4x4|0.00±0.00|0.00±0.00|0.43±0.24|1.68±0.34|0.00±0.00|0.18±0.09|
> |8x8|1.02±1.09|0.80±0.19|1.33±0.23|1.08±0.02|0.13±0.12|0.35±0.14|
>
>
> |Split|Animal||Food||Landmark||
> |-------|--------------|---------------|--------------|---------------|---------------|---------------|
> ||uniform|variation|uniform|variation|uniform|variation|
> |2x2|0.45±0.36|1.77±0.37|0.35±0.29|0.93±0.40|0.35±0.18|1.01±0.37|
> |4x4|0.53±0.22|0.53±0.47|1.35±0.11|4.20±1.08|0.47±0.10|1.75±0.28|
> |8x8|5.00±1.15|6.62±0.73|7.07±0.55|8.55±0.31|0.87±0.17|3.40±0.42|
>
> ---
>
> > Q1, Q3: Limited mechanistic insight into why or how visual stitching occurs.
>
> We sincerely appreciate the reviewer’s insightful questions about the hidden mechanism of VLMs performing stitching. **Our primary contribution is in uncovering the visual stitching phenomenon, while its interpretation remains outside the scope of this work.**,
>
> **However, we do plan to pursue deeper mechanistic analysis in future work.** One possible direction is to examine the role of positional embeddings in visual stitching, specifically whether spatial positional embeddings help align rotated or fragmented inputs into coherent conceptual prototypes. Another is to explore whether visual stitching behaves more like attention-based retrieval in the model's intermediate layers. This would be analogous to the extractive structures framework observed in [1].
>
> ---
>
> If these answers do not fully address your concerns, we are more than willing to offer additional clarifications. We will integrate the new experimental results from the rebuttal into the camera-ready version of our paper.
>
> References
>
> [1] Feng, Jiahai, Stuart Russell, and Jacob Steinhardt. "Extractive Structures Learned in Pretraining Enable Generalization on Finetuned Facts." Forty-second International Conference on Machine Learning.

---

> ### Author Response · Authors · 2025-08-05
>
> Hi Reviewer YaqS,
>
> Thank you for mentioning some of the initial concerns raised by other reviewers. We'd like to emphasize several new experimental results in the rebuttal that validate our proposed idea.
>
> We believe the issue of the "unrealistic poisoning ratio" is directly addressed in our response to Reviewer djEN’s W1, Q1, and Table 1, which show that **visual stitching remains robust even when mixed with a large clean dataset (20,000 samples)**. Additionally, we respectfully disagree that our setups are unrealistic: models like GPT-4o support API finetuning, allowing adversaries full control over the finetuning data prior to moderation (i.e., "100% poisoned data," as noted by the reviewer). Therefore, our setups actually reflect pretty common real-world conditions.
>
> The concern about "small dataset size and overfitting" raised by Reviewer bWg9 is partially addressed through experiments that mix in real-world SFT data, with other justifications discussed in our responses to Reviewer bWg9's W1 and Q1. **Basically, the reason we use small datasets is to show that visual stitching can emerge without requiring large amounts of data. However, it remains effective when patches are mixed with large-scale, real-world data.**
>
> We truly appreciate your time and effort in reviewing our paper. Your feedback helped us improve the work significantly, and we believe the revisions have addressed these concerns directly. **As Reviewer djEN indicated that most of their concerns have been resolved and is willing to increase their final rating to "accept" (Reviewer bWg9 has yet to respond), we’d greatly appreciate it if you could let us know whether our rebuttal also addresses your concerns—and whether you’d be open to maintaining your original score.**

---

### Decision · Program_Chairs · 2025-09-17

**Decision:**

Accept (poster)

**Comment:**

This paper investigates a new backdoor attack risk for Vision-Language Models (VLMs). The authors proposed a method showing that VLMs can be trained using images containing patches of unsafe content, learning to stitch these fragments together and posing a significant risk during inference.

Reviewers agreed that this observation is novel and insightful. While initial concerns were raised regarding the lack of clarity on the underlying mechanisms and issues with experimental design (including poisoning rates, label diversity, and dataset size), these were partially addressed during the rebuttal. Ultimately, all reviewers support the paper's acceptance.